# Oxygen minimum zone cryptic sulfur cycling sustained by offshore transport of key sulfur oxidizing bacteria

Cameron M. Callbeck [1,4], Gaute Lavik[1], Timothy G. Ferdelman [1], Bernhard Fuchs [1], Harald R. Gruber-Vodicka[1], Philipp F. Hach[1], Sten Littmann[1], Niels J. Schoffelen[1], Tim Kalvelage[1], Sören Thomsen[2], Harald Schunck[3], Carolin R. Löscher[3,5], Ruth A. Schmitz [3] & Marcel M.M. Kuypers[1]

Members of the gammaproteobacterial clade SUP05 couple water column sulfide oxidation to nitrate reduction in sulfidic oxygen minimum zones (OMZs). Their abundance in offshore OMZ waters devoid of detectable sulfide has led to the suggestion that local sulfate reduction fuels SUP05-mediated sulfide oxidation in a so-called "cryptic sulfur cycle". We examined the distribution and metabolic capacity of SUP05 in Peru Upwelling waters, using a combination of oceanographic, molecular, biogeochemical and single-cell techniques. A single SUP05 species, *UThioglobus perditus*, was found to be abundant and active in both sulfidic shelf and sulfide-free offshore OMZ waters. Our combined data indicated that mesoscale eddy-driven transport led to the dispersal of *UT. perditus* and elemental sulfur from the sulfidic shelf waters into the offshore OMZ region. This offshore transport of shelf waters provides an alternative explanation for the abundance and activity of sulfide-oxidizing denitrifying bacteria in sulfide-poor offshore OMZ waters.

[1] Max Planck Institute for Marine Microbiology, Bremen D-28359, Germany. [2] GEOMAR Helmholtz Centre for Ocean Research, Kiel D-24148, Germany. [3] Institute for General Microbiology, University of Kiel, Kiel D-24418, Germany. [4] Present address: Swiss Federal Institute of Aquatic Science and Technology (Eawag), Kastanienbaum 6047, Switzerland. [5] Present address: Nordcee and Danish Institute for Advanced Study, Dept. of Biology, University of Southern Denmark, Odense DK-5230, Denmark. Correspondence and requests for materials should be addressed to T.G.F. (email: tferdelm@mpi-bremen.de)

Oxygen minimum zones (OMZs), where dissolved oxygen concentrations fall below 20 µmol kg$^{-1}$, are responsible for large losses of fixed nitrogen from the ocean, despite occupying <1% of the global ocean volume[1–3]. High rates of primary productivity coupled to poor ventilation cause the development of OMZs and can lead to the recurrent accumulation of dissolved hydrogen sulfide in OMZ shelf waters off Peru, Namibia, and India[4–7]. Such sulfidic events often generate episodic plumes of particulate elemental sulfur in surface waters that are visible from space[6, 8–10]. Closely associated with these sulfidic events are bacteria from the gammaproteobacterial clade known as SUP05[5, 6, 11, 12]. As a nitrate-reducing, sulfide-oxidizing chemolithoautotroph, the SUP05 clade encompasses organisms with the physiological capabilities necessary to contribute to the loss of fixed N from productive upwelling regions, the production of climate relevant $N_2O$, and dark carbon fixation in the sub-euphotic water column[5, 6, 11–13]. The extent to which SUP05 organisms are active and directly contribute to dark carbon fixation, sulfide oxidation, and fixed nitrogen loss has not been quantified thus far.

Gene sequences associated with the SUP05 clade, moreover, are frequently found not only in sulfidic shelf waters[5, 6], but in OMZ waters on the outer shelf and offshore OMZ waters where dissolved hydrogen sulfide concentrations fall below typical detection levels (<1 µM)[14–18]. Offshore OMZs also harbor diverse assemblages of putative sulfate-reducing bacteria[13, 15–17], and it has been proposed that SUP05 and sulfate-reducing bacteria may be involved in a "cryptic sulfur cycle"[13]. In this case, cryptic sulfur cycling refers to the simultaneous activity of sulfate-reducing and sulfide-oxidizing pathways in a closely defined space such as a marine particle aggregate. Locally produced sulfide from sulfate-reducing bacteria is immediately oxidized back to elemental sulfur or sulfate by sulfide-oxidizing bacteria and dissolved sulfide remains at sub-micromolar concentrations[13]. Nevertheless, cryptic sulfur cycling may have major implications for nitrogen cycling in offshore OMZ waters. For instance, organic matter mineralization mediated by microbial fermentation coupled to sulfate reduction yields ammonium that can drive anaerobic ammonium oxidation (anammox)[13]. Sulfide oxidation, via nitrate reduction mediated by SUP05, may in turn contribute to the loss of fixed N[13], and to the production of $N_2O$.

SUP05-clade bacteria link nitrogen and sulfur cycling in OMZ water, and several important questions regarding the distribution, metabolic capabilities, and actual activities of SUP05 persist. An accurate census of SUP05 cell abundances in OMZ waters is also absent, in part because the fluorescent in situ hybridization (FISH) probe (GSO477) previously employed to identify SUP05 bacteria targets other sulfide-oxidizing bacteria, for instance, the heterotrophic sulfide-oxidizing Arctic96BD-19 clade[19]. Moreover, the capacity of marine OMZ SUP05 bacteria to perform partial or full denitrification in the ETSP or other marine upwelling ecosystems has also not been determined. A nitrous oxide reductase gene, nosZ, has not been found in SUP05 genomes studied thus far[5, 11, 20], and it has been suggested that other bacteria associated with SUP05 perform the final denitrification step of $N_2O$ reduction to $N_2$[21]. Lastly, despite the persistence of apparent sulfur-based metabolic capacities throughout OMZ waters[13–18], gene sequence abundance cannot be equated with metabolic activity of the corresponding organisms (e.g., activity may diminish in offshore waters). Geochemical evidence that would point to substantial rates of microbial sulfate reduction in offshore waters has not been found in the eastern tropical South Pacific (ETSP), but these natural abundance stable isotope measurements may still be too insensitive to detect estimated rates of sulfur cycling in these OMZ waters[22]. The physical oceanography of the ETSP will play a role in the dispersal of SUP05. In the ETSP region and in other OMZs, mesoscale eddies forming close to the coast are known to facilitate the rapid horizontal advection of coastal geochemical signals and biological communities offshore[23–25]. Thus, the advection of coastal waters into the open ocean may influence the occurrence of SUP05 and sulfate-reducing bacterial communities in offshore waters.

The chemical and hydrographic conditions in the continental Peru Margin waters of the ETSP in austral summer 2013 provided a framework for examining the distribution and activity of SUP05 organisms (Fig. 1a). Under normal flow conditions at the Peru Margin, Ekman transport of the surface, equator-ward flowing Peru Coastal Current results in near-shore upwelling of the oxygen-poor and nutrient-rich water derived from the poleward flowing Peru-Chile Undercurrent[26]. Instabilities in the Peru-Chile Undercurrent possibly triggered by sharp variations in shoreline topography[25] lead to the formation of offshore sub-surface anticyclonic eddies[25, 27–29] (Fig. 1b, c; Supplementary Fig. 1). Thus, in addition to the typical near-shore and offshore ETSP waters, we obtained samples at the offshore site during a period of time when the formation of a sub-surface anticyclonic eddy drove cross-shelf, offshore transport of sulfur-rich shelf waters. To quantitatively discriminate dominant Peru Upwelling SUP05 bacteria from close relatives of the SUP05 clade, we designed and applied a more stringent SUP05 probe. Based on a near-complete metagenomics bin we reconstructed the metabolic capabilities of the Peru Upwelling SUP05 bacteria. Finally, we specifically determined the single-cell C uptake activity of SUP05 bacteria via isotope-labeling experiments combined with nanoscale secondary ion mass spectrometry (nanoSIMS) analysis. This allowed us to evaluate the contribution of SUP05 activity to carbon, nitrogen, and sulfur cycling both in near-shore and in offshore ETSP OMZ waters. We provide direct data on the in situ activity of a sulfide-oxidizing, nitrate-reducing organism of importance in OMZ waters, and find that SUP05 continues to be active in waters that are transported offshore along with elemental sulfur.

## Results

**Biogeochemical characterization of shelf and offshore waters.**
Waters from the ETSP region off the coast of Peru (12°S 78.5°W and 13.5°S 77°W) were sampled from 8 February to 4 March 2013 onboard the RV *Meteor* (Expedition M93; Supplementary Table 1). At the beginning of the sampling period in February 2013, an anticyclonic mesocale eddy had formed approximately 50 km from the coast (Fig. 1b; Supplementary Fig. 1b). During the course of our experiments and sampling, the sub-surface eddy expanded and propagated in south-southwesterly direction, eventually veering off in a westerly direction (Fig. 1b; S1c–f). By March 2013, the eddy had caused a filament of surface shelf water moving along the northern rim of the eddy to extend to nearly 330 km offshore (Fig. 1b). Sub-surface waters at stations U2, U3, L1, and L3 were impacted by the resulting cross-shelf transport of shelf waters during and after the eddy formation[25]. Station L2, on the other hand, was sampled after the eddy had traveled already further westwards and caused the onshore advection of offshore water masses along its southern rim[25]. Thus, station L2, referred to here as "non-eddy" for simplicity, exhibited temperature–salinity characteristics typical of offshore waters, which are clearly separated in the temperature and salinity space from the stations impacted by coastal waters (Supplementary Fig. 2).

Station U1 on the shelf was also sampled in early March when normal (non-eddy) flow conditions prevailed (Supplementary Fig. 1f). The near-shore, shelf waters at station U1 were characterized by extreme depletion of dissolved oxygen (below 10 m) and nitrate (below 30 m), and the presence of free dissolved

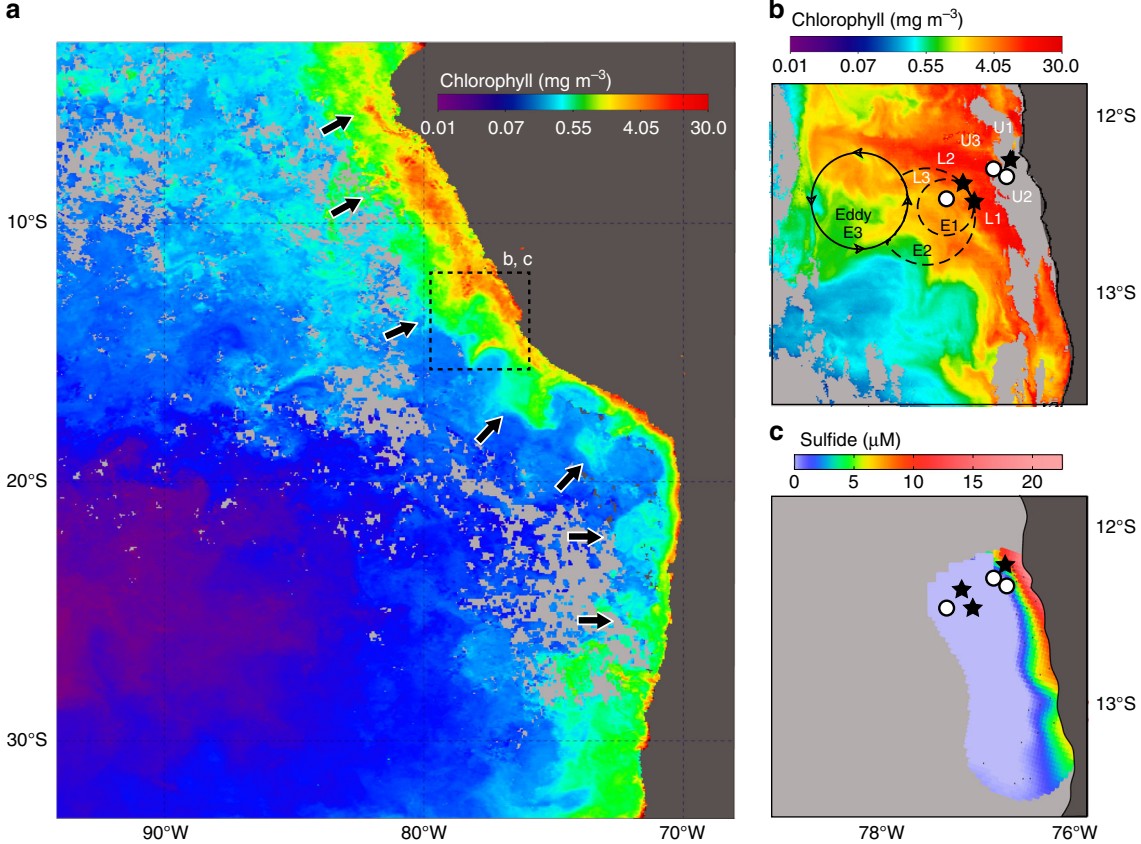

**Fig. 1** Station and mesoscale eddy location relative to near-surface chlorophyll a and maximum dissolved sulfide concentrations. **a** Monthly composite MODIS image (see Methods for source) showing near-surface chlorophyll concentrations for March 2013, where the arrows indicate cross-shelf advected filaments. **b** MODIS image of near-surface chlorophyll concentrations for 24 February 2013. The main water column sampling stations (U1, L1, and L2) are marked with black stars; additional stations with white circles. Times of station sampling are provided in Supplementary Table 1. Formation and propagation of the eddy westward occurring over time is indicated: E1 represents the initial eddy formation from 28 January to 3 February; E2 shows the expansion of the eddy (7–12 February 2013); and E3 is the location of the eddy when the image was taken (24 February 2013). **c** Maximum sulfide concentration reported for water masses with densities between 26.1 and 26.2 kg m$^{-3}$

hydrogen sulfide (up to 7 μM) and ammonium (up to 6 μM) (Fig. 2; Supplementary Figs. 3, 4). Nitrate-depleted, sulfide and elemental sulfur-rich bottom waters covered the entire near-shore Peruvian shelf between 12°S 78.3°W and 13.3°S 77°W (Fig. 1c; Supplementary Fig. 3). The reduced sulfur inventories in February–March 2013 ($1.6 \times 10^9$ moles $H_2S$ and $7.0 \times 10^8$ moles elemental sulfur) were more than twice as large as for the sulfidic event reported for the same area in 2009[5].

A nitrate-sulfide chemocline in the inner shelf waters at 25–35 m water depth (hereafter simply referred to as the chemocline) coincided with peaks of nitrite and elemental sulfur (Fig. 2; Supplementary Fig. 4). Elemental sulfur, or cyclooctasulfur $S_8$, is highly insoluble in sea water[30]. The elemental sulfur measured in OMZ waters likely exists as colloidal sulfur; $S_8$ externally associated with particles and cells; internal cellular deposits of $S_8$; as well as the sulfane component of dissolved inorganic polysulfides. An intermediate product of biotic and abiotic sulfide oxidation, elemental sulfur, reached concentrations of up to 6 μM within the chemocline and persisted at μM concentrations in the deeper, sulfidic waters, most likely as inorganic polysulfides (Fig. 2; Supplementary Fig. 4). Elemental sulfur likely formed at 30–35 m as chemolithotrophic organisms used downward mixed nitrate to oxidize hydrogen sulfide. Under the denitrifying conditions found at the base of the chemocline, elemental sulfur is the first product of sulfide oxidation[31] as depicted in Eq. 1:

$$5H_2S + 2NO_3^- + 2H^+ \rightarrow 5S^0 + N_2 + 6H_2O \qquad (1)$$

Elemental sulfur, which is transported through eddy diffusion throughout the chemocline, may fuel further nitrate consumption via denitrification as shown in Eq. 2.

$$5S^0 + 6NO_3^- + 2H_2O \rightarrow 5SO_4^{-2} + 3N_2 + 4H^+ \qquad (2)$$

Overall, as estimated from nitrate, elemental sulfur, and sulfide concentration gradients, and employing an eddy diffusion coefficient of $1.4 \times 10^{-4}$ m$^2$ s$^{-1}$ (see Methods), the downward nitrate flux into the chemocline (17 mmol S m$^{-2}$ d$^{-1}$) was more than sufficient to oxidize the upward flux of sulfide (−7.6 mmol S m$^{-2}$ d$^{-1}$) completely to sulfate via denitrification (combined Eqs. 1 and 2 as shown in Eq. 3)

$$5H_2S + 8NO_3^- \rightarrow 5SO_4^{-2} + 4N_2 + 4H_2O + 2H^+ \qquad (3)$$

These estimates encompass the upward eddy diffusion flux of elemental sulfur in the chemocline, an upper limit for which we estimate to be −6.6 mmol S m$^{-2}$ d$^{-1}$. Up to 70% of the total nitrate flux could be attributed to the oxidation of sulfide within the chemocline at station U1. Microorganisms such as SUP05 that can couple dissolved sulfide oxidation to nitrate reduction should, therefore, dominate this interface between deep sulfidic waters and overlying nitrate.

**Factors controlling Peru Upwelling SUP05 distributions.** The same species-level clade of SUP05 bacteria were identified based

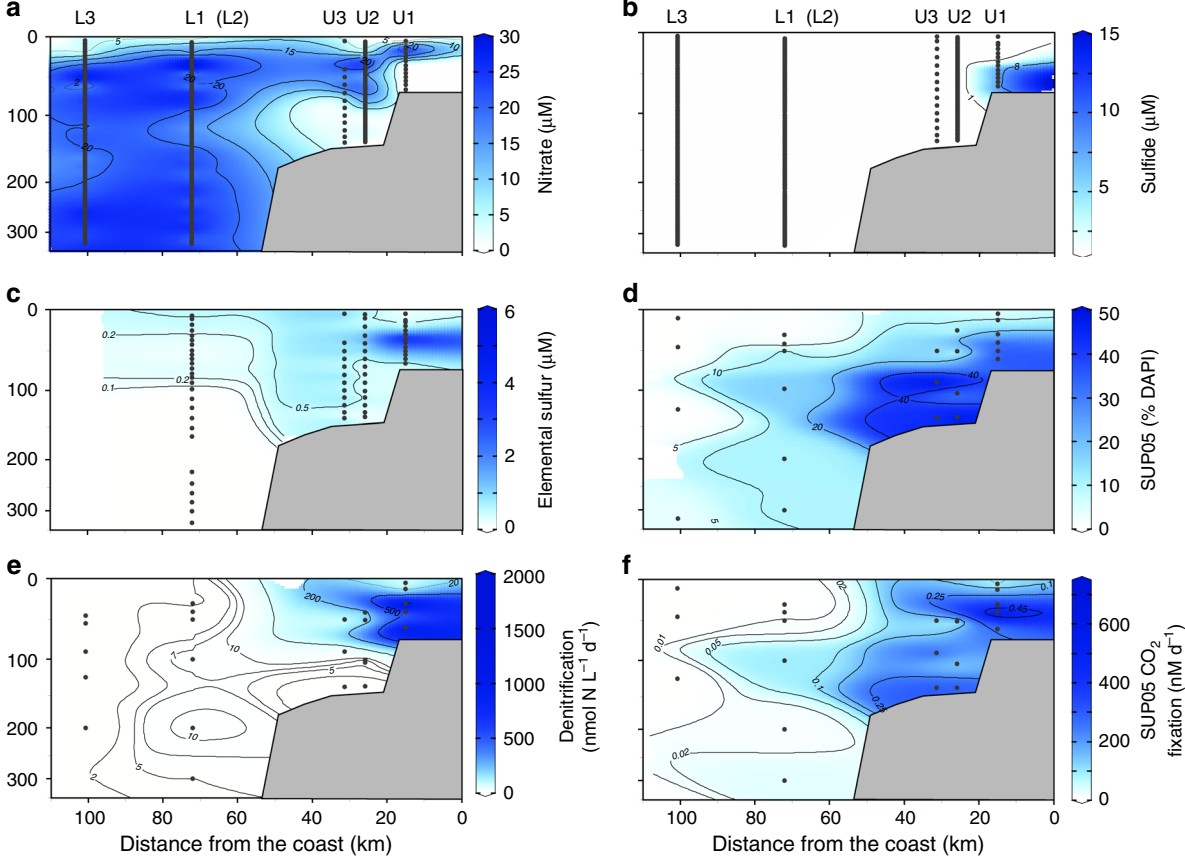

**Fig. 2** Distribution of concentrations, abundances, and bulk and single-cell activities in the Peru Upwelling OMZ as a function of distance from the coast. The composite plots show depth and cross-shelf distribution **a** nitrate, **b** dissolved sulfide, **c** elemental sulfur, **d** % total bacteria (DAPI) identified as SUP05, **e** bulk rates of denitrification, and **f** single-cell determined rates of $CO_2$ fixed by SUP05 at the time of eddy-induced offshore transport of shelf waters. Note that station L2 (not included in composite) was located near station L1, but occupied 11 days later. Black dots indicate sample depths at each station included in the composite plots

on 16S rRNA gene analysis, in both sulfidic (stations U1 and U1a) and in the sulfide-free offshore waters (e.g., station L2). A new FISH probe, GSO131, was designed to target this group; the probe's high specificity clearly distinguish Peru Upwelling SUP05-clade bacteria from near relatives within the Gammaproteobacterial sulfide oxidizer (GSO) clade (e.g., Arctic96BD-19 bacteria; Fig. 3; Supplementary Tables 2, 3; see also discussion on specificity of GSO131 in Supplementary Discussion). Peru Upwelling SUP05 bacteria as quantified using the GSO131 probe composed up to 50% ($1.7–3.2 \times 10^6$ cells per mL) of the total microbial community within the chemocline at station U1 (Fig. 2d; Supplementary Fig. 4). Similar cell densities using a less-specific SUP05 FISH probe have been reported for the Namibian shelf region where sulfidic conditions prevailed[6]. At station U1, peak SUP05 cell densities within the chemocline coincided with peak rates of denitrification ($2000 \text{ nmol N L}^{-1} \text{ d}^{-1}$; Fig. 2e; Supplementary Fig. 4) and dark carbon fixation ($600–1000 \text{ nmol C L}^{-1} \text{ d}^{-1}$; Supplementary Fig. 4). These results support earlier conclusions that SUP05 is a dominant taxon mediating sulfide-driven denitrification at the chemocline in such sulfidic, upwelling shelf waters[5, 6].

In contrast to the sulfide-rich, nitrate-deplete waters on the inner shelf, total dissolved sulfide concentrations dropped below detection ($<1 \mu M$) on the outer shelf (stations U2 and U3; Fig. 2b) and offshore beyond the outer shelf break. Elemental sulfur remained detectable at 100–1000 nM in the oxygen- and nitrate-depleted deep waters of outer shelf stations U2 and U3, and more

interestingly, persisted in the offshore eddy-influenced waters of station L1 where dissolved sulfide was not detected (Fig. 2; Supplementary Fig. 4). At station L1, elemental sulfur concentrations ranged between 50 and 750 nM from 5 to 100 m depth, and coincided with a nitrate minimum (Fig. 2; Supplementary Fig. 4). Temperature–salinity properties at station L1 suggest that nitrate-depleted coastal waters containing elemental sulfur were transported offshore along isopycnals up to 80 km from the coast (Fig. 2a, c; Supplementary Fig. 2). After the eddy had moved further offshore, elemental sulfur was restricted to a narrow band near the surface with concentrations of $<300$ nM as seen at station L2 (Fig. 2; Supplementary Figs. 2 and 4).

At the eddy-influenced offshore station L1, SUP05 cell densities of up to $4.5 \times 10^5$ cells per mL comprised a significant fraction (up to 17%) of the microbial community even in the absence of dissolved sulfide (Fig. 2; Supplementary Fig. 4). Rates of nitrate reduction to $N_2$ at L1 ranged from 5 to 16 $\text{nmol N L}^{-1} \text{d}^{-1}$, and dark carbon fixation of 9–130 $\text{nmol C L}^{-1} \text{d}^{-1}$ (Fig. 2e; Supplementary Fig. 4) were much lower than those at station U1 on the inner shelf. Nonetheless, rates of denitrification and dark carbon fixation at station L1 under the influence of the eddy and cross-shelf transport exceeded those observed under "non-eddy" conditions (L2). At station L2 we measured reduced rates of denitrification ($<0.13–4.3$ $\text{nmol N L}^{-1} \text{d}^{-1}$) and dark carbon fixation (11 to 51 $\text{nmol C L}^{-1} \text{d}^{-1}$; Fig. 2e; Supplementary Fig. 4). Correspondingly, SUP05 abundances at L2 were one order of magnitude lower than at station L1 and comprised only a minor fraction (0–2%) of the

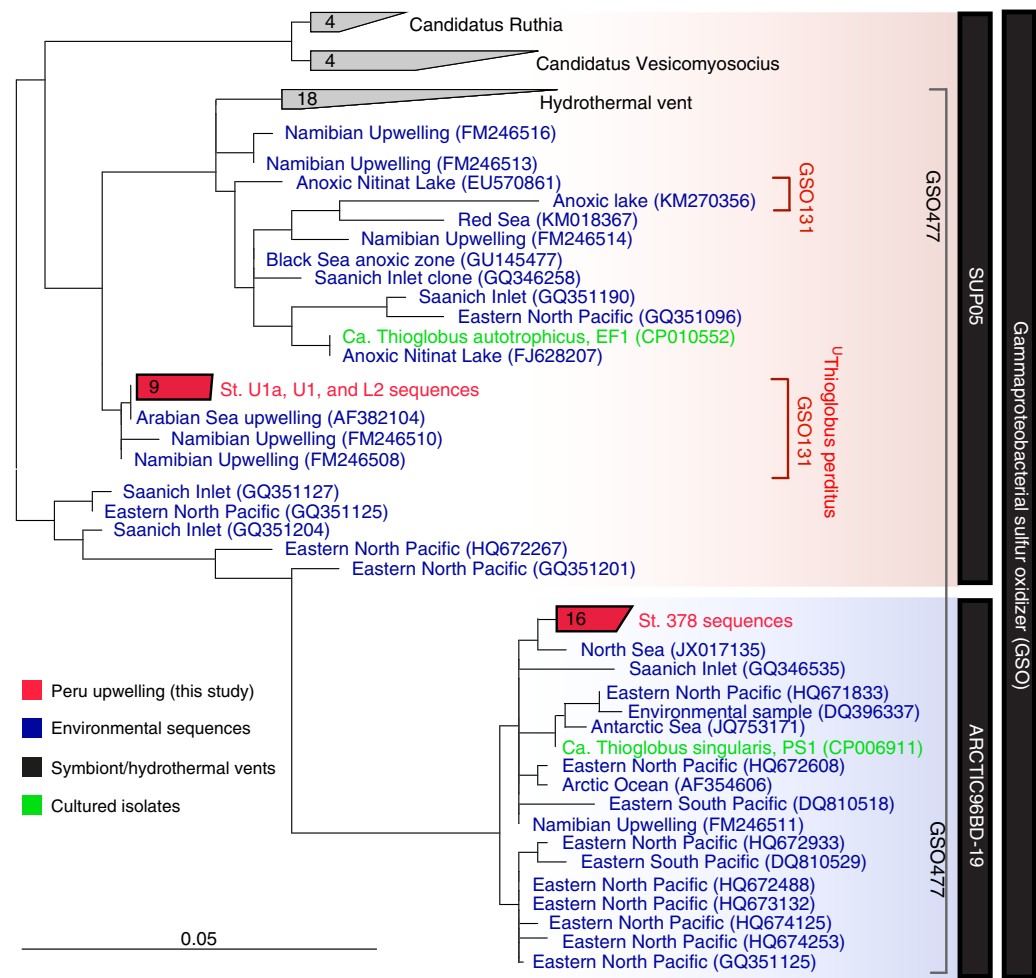

**Fig. 3** Phylogenetic diversity of GSO 16S rRNA genes recovered from sulfidic and non-sulfidic stations from the Peruvian upwelling region. The phylogenetic tree was calculated using the neighbor joining and RAxML methods, including various filters, an unrooted consensus tree is shown. The typeface in blue, black, and green represent sequences recovered from other studies. Typeface in red represents sequences recovered from sulfidic stations U1 and U1a, and from offshore stations L1, L2, and 378 (Supplementary Table 1). The coverage and specificity of the newly designed FISH GSO131 probe is indicated by the red brackets; for overall probe coverage details see Supplementary Table 3. The broad coverage GSO477 probe is indicated by the gray bracket

microbial community. Denitrification rates also broadly correlated with SUP05 cell densities ($DN = (1.14 \times \log[\text{cell per L}]) - 7.88$; $R^2 = 0.71$).

It has been proposed that sulfide produced via microbial sulfate reduction in marine particle aggregates in the offshore stations may also fuel offshore SUP05-mediated nitrate reduction, i.e., a cryptic sulfur cycle[13]. We observed aggregates containing deltaproteobacteria (e.g., sulfate-reducing bacteria) in addition to SUP05 in the samples at the shelf water-influenced offshore station L1 (Supplementary Fig. 5). Depth-integrated SUP05 abundances, however, greatly exceeded delta-proteobacteria by nearly sevenfold (Supplementary Fig. 5), which is also consistent with other metagenomic and functional gene surveys of offshore OMZ waters that find that key sulfur-based genes affiliated to sulfide-oxidizing bacteria consistently outnumber genes affiliated to sulfate-reducing bacteria[16, 17]. We cannot discount sulfide production from sulfate-reducing bacteria co-transported with SUP05. Nonetheless, given rates of sulfide oxidation ostensibly associated with a cryptic sulfur cycle[13] (single-cell discussion below), the SUP05 bacteria could continue for days to weeks to metabolize the large amounts of elemental sulfur transported offshore (50–175 nmol $L^{-1}$ or >20 mmol $m^{-2}$ at station L1, Supplementary Figs. 3, 4).

**Single-cell activities of SUP05 bacteria**. The presence and abundance of an organism in any given environment, for instance SUP05 distributions in eddy-influenced offshore waters, yields only limited information on the activity of the organism and its actual impact on the chemistry of the environment. To address the impact of SUP05 on the cycling and fate of carbon, sulfur, and nitrogen in ETSP waters, we compared the SUP05-specific carbon assimilation in the chemocline at station U1, where SUP05 likely plays a dominant role in coupling sulfide oxidation with denitrification, with SUP05-specific activities at the "eddy-influenced" offshore (L1) and "non-eddy" (L2) stations. We quantified the specific contribution of SUP05 bacteria to dark carbon fixation by measuring the assimilation of $^{13}$C-bicarbonate into SUP05 biomass at the single-cell level using nanoSIMS technology. In experiments from station U1 with close to ambient concentrations of sulfide, elemental sulfur, and nitrate, SUP05 fixed $CO_2$ at a cell specific rate of $0.19 \pm 0.02$ fmol C per cell per day (averaged from 30 and 60 m depths, Fig. 4; Table 1). SUP05-specific $^{15}$N-$NO_3^-$ assimilation rates also increased linearly with SUP05 carbon assimilation rates in the same experiments (Supplementary Fig. 6). At station L1, the SUP05 $CO_2$ fixation rate of $0.17 \pm 0.02$ fmol C per cell per day was similar to the specific $CO_2$ fixation rate determined at station U1 (Fig. 4a; analysis of variance

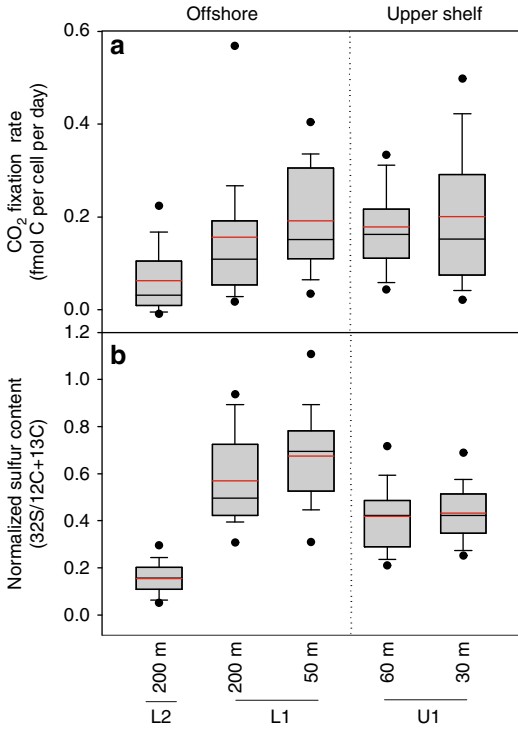

**Fig. 4** SUP05 single-cell activity and sulfur content of ETSP-SUP05 bacteria. **a** $CO_2$ fixation rates based on $^{13}$C-bicarbonate uptake into SUP05 cells. **b** Normalized single-cell sulfur content. The mean (red line) and median (black line) are indicated. The boxes represent the distribution of data with 95th and 5th percentiles and outliers are indicated by the black circles. Standard deviations bars are shown. The number of SUP05 cells analyzed at station–depths were as follows: U1–30 m (48 cells); U1–60 m (59 cells); L1–50 m (35 cells); L1–200 m (32 cells); and L2–200 m (23 cells)

(ANOVA), $p = 0.14$, no statistical difference). In the offshore waters that were unaffected by the eddy, station L2, SUP05 bacteria fixed C at a substantially lower rate of $0.06 \pm 0.01$ fmol C per cell per day rates (Fig. 4a; ANOVA, $p = <0.001$). Thus, the uptake of $^{13}CO_2$ as determined at the single-cell level showed that SUP05 bacteria actively assimilated $CO_2$ in shelf waters, as well as in shelf waters that had recently been transported offshore. In offshore ETSP water masses isolated from the shelf, however, SUP05 bacteria were much less active.

We can put the activity of SUP05 cells into perspective with respect to carbon, nitrogen, and sulfur cycling in the ETSP waters by upscaling the SUP05-specific carbon uptake rates. There are large uncertainties inherent to these calculations, as with most upscaling or extrapolations. The C uptake rate for SUP05 is dependent on the cell size and carbon density of growing SUP05 cells. Carbon content and carbon density of living bacterial and archeal cells are points of considerable discussion[32], and will dictate calculated $CO_2$ fixation rates (see Eq. 7 in Methods). We estimated the C content of the SUP05 cells analyzed by nanoSIMS using the power relationship based on the most recent cumulative data for bacterial sizes and biomasses[32]. SUP05 cells in the Peru Upwelling at the time of sampling were relatively large (0.18–0.39 $\mu m^3$ corresponding to 61–83 fg C per cell; Table 1). SUP05 contributed substantially to the total bacterial carbon and dark carbon fixation rates (Table 1; Fig. 2f) at stations U1 and L1, which is in line with their large abundances and high per cell C assimilation rates at these stations. Our estimates suggest that SUP05 bacteria are on average responsible for up to 50% of the dark carbon fixation rates on the shelf. They are consistent with

the observation that there are also other chemolithoautotrophic organisms present in the shelf waters capable of denitrification and sulfide oxidation (e.g., Epsilonproteobacteria). At the offshore eddy-influenced L1 station, most of the dark carbon fixation could be attributed to SUP05 (up to 100%), whereas only 7% of the dark carbon fixation could be attributed to SUP05 at the non-eddy-influenced station L2 (Table 1). (These are upper values for % dark carbon fixation, because the bulk dark carbon fixation rates may be underestimated as the nominal size of the glass fiber filters used for the bulk $^{13}CO_2$ uptake determinations is larger than the 0.2 µm pore size of the polycarbonate (PC) filters used for the single-cell $^{13}CO_2$ uptake rates.) It is interesting to note that Hawley et al.[12], in extrapolating from a gene-centric biogeochemical model based on quantitative polymerase chain reaction (qPCR) data and the abundances of SUP05 C fixation proteins in Saanich Inlet, estimated SUP05-associated C fixation rates of 10–120 nmol C L$^{-1}$ d$^{-1}$ for OMZ waters. Our calculated in situ rates of SUP05-specific carbon fixation rates for the ETSP OMZ waters (ranging from 1.3 to 592 nmol C L$^{-1}$ d$^{-1}$ for stations U1 and L1; Table 1) confirm the proposed importance of SUP05 to dark carbon fixation in shelf and sulfur-rich OMZ waters.

Growing, or actively autotrophic SUP05 cells would also be expected to have an impact on denitrification and sulfide oxidation. In contrast to the determination of SUP05-specific C or N assimilation using nanoSIMS, the direct experimental determination of single-cell or single-clade respiration rates in the environment is not yet possible. We can, however, estimate the potential impact of the SUP05 bacteria nitrogen cycling in the offshore eddy- and non-eddy-influenced waters by assuming that the biomass yield, i.e., the amount of $CO_2$ fixed per cell SUP05 per mol nitrate reduced or per mol sulfide or sulfur oxidized is similar for both near-shore and offshore SUP05 cells. Cultivated sulfide oxidizers growing on sulfide and oxygen exhibit biomass yields of 0.35–0.58 mol $CO_2$ fixed per mol $H_2S$ oxidized[33–35]. Assuming similar yields for nitrate-dependent sulfide oxidation yields 0.22–0.37 mol $CO_2$ fixed per mol nitrate reduced based on the stoichiometry in Eq. 3. Estimated rates of sulfide-dependent denitrification attributable to SUP05 at stations U1 (830–2180 nmol N L$^{-1}$ d$^{-1}$) and L2 (1–3 nmol N L$^{-1}$ d$^{-1}$) are at the upper range of experimentally determined bulk denitrification rates (Table 1), or exceed bulk denitrification rates by up to threefold at station L1 (24–62 nmol N L$^{-1}$ d$^{-1}$). Estimates of nitrate-dependent sulfur oxidation in the sulfur-rich, offshore waters at station L1 (15–40 nmol S L$^{-1}$ d$^{-1}$) are approximately two- to threefold greater than sulfide oxidation rates measured by Canfield et al.[13] of 5–21 nmol L$^{-1}$ d$^{-1}$. All of these estimates involve large uncertainties. Environmental biomass yields for important chemolithoautotrophic denitrifying processes are lacking and are likely to be lower than those determined for aerobic processes and pure culture organisms[36]. SUP05 may also perform aerobic sulfur oxidation[5, 11] (see also the discussion in the next section), and thus, part of its C fixation activity may be linked to microaerophilic respiration, especially in offshore waters. Nevertheless, the high rates of $CO_2$-fixing activity directly linked to SUP05 cells suggest that SUP05 has the potential to substantially contribute to rates of denitrification and sulfur oxidation both on the shelf and in sulfur-rich shelf waters transported offshore.

**Peru Upwelling SUP05 ecophysiology**. Metagenomics, in combination with nanoSIMS analysis, show that SUP05 is well adapted to the sulfide-poor conditions in water masses transported offshore. We assembled and binned a draft genome at 95% completeness based on Gammaproteobacterial marker genes of

**Table 1 Single-cell SUP05 CO₂ assimilation rates, contribution to dark CO₂-fixation, and bulk nitrate reduction rates**

|  | Station U1 | Station L1 | Station L2 |
|---|---|---|---|
| SUP05 cell abundance (cells per L) |  |  |  |
| Range | $1.1\text{-}9.1 \times 10^9$ | $0.18\text{-}4.4 \times 10^8$ | $0.29\text{-}2.9 \times 10^7$ |
| Mean | $1.7 \times 10^9$ | $1.3 \times 10^8$ | $1.2 \times 10^7$ |
| *Single-cell carbon fixation rates*[a] |  |  |  |
| Number of SUP05 cells analyzed | 107 | 67 | 23 |
| Cell volume ($\mu m^3$) |  |  |  |
| Mean | $0.18 \pm 0.01$ | $0.39 \pm 0.04$ | $0.31 \pm 0.06$ |
| Cell carbon content (fmol C per cell) |  |  |  |
| Mean | $5.0 \pm 0.3$ | $6.9 \pm 0.7$ | $6.1 \pm 1.2$ |
| C turnover (per day) |  |  |  |
| Mean | $0.038 \pm 0.03$ | $0.025 \pm .03$ | $0.010 \pm 0.02$ |
| Per-cell fixation rate (fmol C per cell per day) |  |  |  |
| Mean | $0.19 \pm 0.02$ | $0.17 \pm 0.02$ | $0.06 \pm 0.01$ |
| *SUP05 contribution to CO₂ fixation*[b,c] |  |  |  |
| Volumetric SUP05 $CO_2$ fixation rates (nmol C $L^{-1} d^{-1}$) |  |  |  |
| Range | 171–592 | 1.3–77 | 0–5.1 |
| Mean | 324 | 22 | 0.7 |
| Depth-integrated SUP05 $CO_2$ fixation rate (mmol C $m^{-2} d^{-1}$) | $13.4 \pm 1.0$ | $8.4 \pm 1.0$ | $0.32 \pm 0.07$ |
| Bulk depth-integrated dark $CO_2$ fixation (mmol C $m^{-2} d^{-1}$) | $26.2 \pm 2.0$ | $8.0 \pm 1.0$ | $4.5 \pm 0.6$ |
| % Dark $CO_2$ fixation by SUP05 | 51 | 100 | 7 |
| *Measured bulk nitrate reduction rates* |  |  |  |
| Dentrification to $N_2$ (nmol N $L^{-1} d^{-1}$) |  |  |  |
| Range | 0–2044 | 0–15.9 | 0–4.3 |
| Mean | 1078 | 7.9 | 0.7 |
| $NO_3^-$ to $NO_2^-$ reduction (nmol N $L^{-1} d^{-1}$) |  |  |  |
| Range | NA | 10.3–24.8 | 0–16.2 |
| Mean |  | 17.9 | 7.7 |

See Methods for details regarding single-cell calculations
[a] Rates and abundances are from stations and depths, where nanoSIMS measurements were performed: station U1 (30 and 60 m); station L1 (50 and 200 m); and station L2 (200 m)
[b] Calculations were made using the per-cell SUP05 $CO_2$ fixation rate multiplied through the range of measured SUP05 cell abundances determined in the OMZ waters
[c] Dark $CO_2$ fixation rates integrated for station U1 over 30–65 m, L1 over 100–300 m, and L2 over 125–320 m

the Peru Upwelling SUP05 from the metagenome for station U1. The draft genome encoded genes involved in the reverse dissimilatory sulfite reduction pathway (*rdsr*) used in the oxidation of intracellular $S^0$, as well as an incomplete periplasmic thiosulfate oxidation pathway by *sox* (Fig. 5). The incomplete sox pathway, specifically the absence of *soxCD* genes, has previously been observed for members of the SUP05 bacterial clade[5, 11, 37, 38], and has been linked to intracellular sulfur deposits in other sulfide oxidizers[39, 40]. Empirically, sulfur deposits have been shown to accumulate intracellularly in Arctic96BD-19 bacteria, a closely related lineage of SUP05[41]. Moreover, with nanoSIMS, we found that SUP05 cells at stations U1 and L1 had significantly greater (ANOVA, $p = <0.001$) sulfur content compared to cells at station L2 (Fig. 4b), which suggested that SUP05 has a capacity to store and consume sulfur. The stored sulfur, deposited in an inorganic or organic form, is putatively oxidized via the rdsr pathway[42]. The organism can also use thiosulfate. Thiosulfate was detected in the chemocline at station U1 (400 nM), but was below detection (50 nM) at offshore stations L1 and L2 (Supplementary Fig. 4). The presence of the high-affinity cytochrome cbb3 complex indicates that electrons from the oxidation of reduced sulfur compounds can be used to reduce trace concentrations of dissolved $O_2$ to water. The obtained ETSP-SUP05 genome shows that energy can also be conserved by coupling the oxidation of sulfur to nitrate reduction to $N_2$ (Fig. 5). In contrast to the Saanich Inlet SUP05 metagenome that lacks the nitrous oxide reductase (*nosZ*) gene[11, 12, 20], the ETSP-SUP05 genome encodes a complete denitrification pathway, including *nosZ* (Fig. 5). The detection of the SUP05 *nosZ* gene at the same coverage as the rest of the SUP05 genome in stringent metagenomic read mappings at station L1 indicate that, functionally, the same SUP05-clade strain was present as in station U1. Thus, the genomic capability of

SUP05 organisms active in the ETSP predicts that they can perform complete denitrification coupled to sulfide and elemental sulfur oxidation.

The Peru Upwelling SUP05 bacteria described here has only a 97.6% 16S rRNA sequence identity (ANI percentage of 74%) with *Candidatus Thioglobus autotrophicus*[20]. Unlike *Ca. T. autotrophicus*, it has the full denitrification pathway. As we are able to distinguish the Peru Upwelling SUP05 clade at the species level[43], we propose a uncultivated taxa (U) name for the Peru Upwelling SUP05 bacteria "*UThioglobus perditus*"[44]. *Perditus* means lost. The Peru Upwelling SUP05 bacterium $^U$*T. perditus* finds itself lost in the offshore OMZ waters.

**Implications for cryptic sulfur cycling.** Mesoscale eddies, like the one described in this study, are common throughout the ETSP OMZ. Long-term remote sensing data indicate that ~50% of the ETSP OMZ area is covered by mesoscale eddies at any one time[45] that likely have a substantial impact on the chemistry and biology of offshore waters[46]. In March 2013, several chlorophyll-rich filaments were seen projecting from the ETSP coastline with some of the filaments extending up to a remarkable 1500 km from the coast (Fig. 1 and Supplementary Fig. 7). Our results show that in addition to chlorophyll, reduced sulfur from the anoxic inner shelf waters was transported offshore as a result of mesoscale processes. During our research campaign $1.6 \times 10^9$ moles of dissolved sulfide $H_2S$ and $7.0 \times 10^8$ moles of elemental sulfur accumulated on the inner shelf. We conclude that a substantial part of this reduced sulfur escaped oxidation on the shelf and was transported offshore. Moreover, mesoscale eddy-driven cross-shelf transport led to the dispersal of microbes such as the SUP05-clade bacteria $^U$*T. perditus* from the sulfidic shelf waters to the offshore OMZ waters. The single-cell carbon fixation data

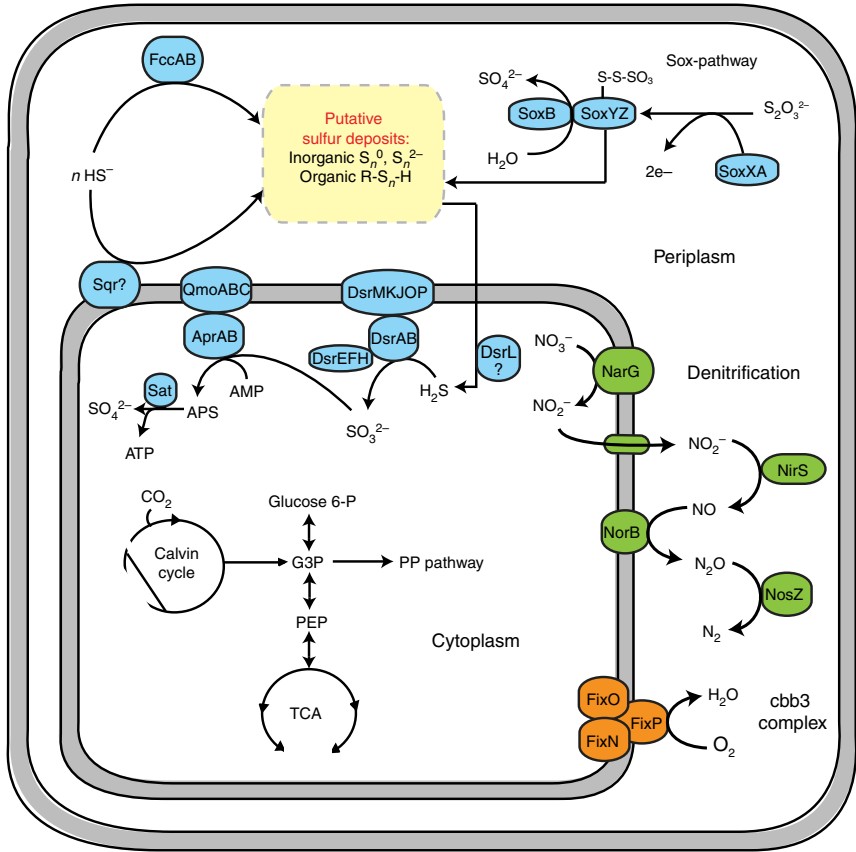

**Fig. 5** Key metabolic pathways encoded in a SUP05 $^{U}$Thioglobus perditus population genome bin: Nar, nitrate reductase; Nir, nitrite reductase; Nor, nitric oxide reductase; Nos, nitrous oxide reductase; Dsr, dissmilatory sulfite reductase; Apr, adenylylsulfate reductase; Sat, sulfate adenylyltransferase; Fcc, sulfide-binding flavoprotein; Sqr, sulfide-quinone reductase. The metabolic prediction is based on a 95% complete SUP05 draft genome recovered from station U1. For a complete list of genes please refer to Supplementary Table 4

show that the populations of $^{U}$T. perditus continued to be highly active in these non-sulfidic offshore OMZ waters.

Our results show that mesoscale eddy-driven water mass movement can explain the abundance and activity of sulfide-oxidizing denitrifying bacteria in sulfide-poor offshore OMZ waters and can drive "cryptic sulfur cycling" via the continued oxidation of co-transported elemental sulfur. In fact, chlorophyll-rich filaments indicating cross-shelf transport also occurs in the Chilean upwelling region where cryptic sulfur cycling was first reported[13] (e.g., Supplementary Fig. 7). Thus, eddy-driven cross-shelf transport combined with the capability of $^{U}$T. perditus to denitrify and thrive on elemental sulfur in the absence of dissolved sulfide may contribute to its success in OMZ waters worldwide.

## Methods

**Sampling and hydrography**. Waters were sampled in the ETSP region off the coast of Peru (12°S 78.5°W and 13.5°S 77°W) from 8 February to 4 March 2013 onboard the RV *Meteor* (Supplementary Table 1). Either a CTD rosette equipped with twenty-four 10 L Niskin bottles was used to collect water samples or a pump-CTD. Oxygen, temperature, and salinity were recorded with depth on both up and downcasts of the CTD. The mesoscale eddy and shelf currents were tracked by horizontal velocities surveyed by glider deployments and vessel mounted acoustic doppler current profilers from January to March 2013[25].

**Nutrient and sulfur chemistry**. Samples for nutrient and reduced sulfur chemistry were obtained from the pump-CTD downcasts. Sulfide concentrations were determined by the methylene blue method[47] immediately from Niskin bottles using 4 mL of sample and 320 μL of diamine reagent. The diamine solution and samples were incubated in the dark at ambient temperatures (18–22 °C) prior to measuring with a spectrophotometry at 670 nm. The detection limit of this method is 1 μM. Separate nutrient samples were taken for the analysis of nitrate, nitrite, and

ammonium, and were measured onboard with a QuAAtro autoanalyzer (Seal Analytical). The detection limits are 0.1, 0.1, and 0.3 μM, respectively.

For the analysis of elemental sulfur chemistry sulfidic waters were immediately fixed in zinc chloride (100 μL of 20% (weight/weight) in 50 mL sample), and stored at −20 °C. Elemental sulfur was extracted by a chloroform-methanol procedure using 5–15 mL of sample volume[48]. Internal standard (31.2 mg of 4,4′-dibromo diphenyl (Sigma Aldrich)) dissolved in 100 mL methanol) was added to back calculate the extraction efficiency. Three rounds of chloroform extraction (500 μL each) were performed. After each step the chloroform sample mixture was sonicated for 15 min at 4 °C and then the chloroform was pipetted off into a glass vial where it was concentrated under an $N_2$ stream. In the last stage the extracted product was dissolved in methanol and filtered to remove larger particles (0.45 μm filter). The methanol-dissolved sample was measured by ultrahigh pressure liquid chromatography (UPLC) using a Waters Acquity H-class instrument with a Waters column (Aquity UPLC BEH C18, 1.7 μm, 2.1 × 50 mm column; Waters, Japan) and methanol eluent flowing at 0.4 mL min$^{-1}$ equipped with a Waters PDA detector (absorbance wavelength set to 265 nm). The detection limit of elemental sulfur using this method was 50 nM.

Thiosulfate concentrations in collected seawater samples were quantified by the monobromobimane derivatization method[49]. Derivatization of the samples was done in 1.8 mL glass vials in a dark room. A 500 μL sample was fixed in 50 μL of HEPES-EDTA buffer (pH 8) and 50 μL of monobromobimane (45–48 mM, Sigma Aldrich). The reaction was stopped after 30 min by adding 50 μL of methansulfonic acid (324 mM). Bimane fixed samples were measured by UPLC using a Waters Acquity H-class instrument with a Waters column (Aquity UPLC BEH C8, 1.7 μm, 2.1 × 50 mm column; Waters, Japan) and an acetic acid/methanol gradient flowing at 0.65 mL min$^{-1}$ equipped with a Waters FLR detector (excitation and absorbance wavelength was set to 380 and 480 nm, respectively). The detection limit of this method based on standard preparations was 50 nM. Integration of peak areas was done using Waters Empower 3 software.

The sulfide, sulfur, and nitrate fluxes were determined at the chemocline at station U1 from 30 to 40 m, 20 to 30 m, and 12 to 30 m depth, respectively. The eddy diffusivity (1.4 × 10$^{-4}$ m$^2$ s$^{-1}$) was determined for the mid to upper shelf of the Peruvian upwelling region from microstructure profiles (Schlosser et al., in preparation). A negative value indicates an upward water column flux.

**15N- and 13C-labeled incubation experiments**. Sea water was transferred from the Niskin bottles into 250 mL glass serum bottles and $^{15}$N-labeled incubation experiments were performed according to Holtappels et al.[50]. Bottles were allowed to gently overflow two to three times and then were capped avoiding oxygen contamination. All bottles, unless sampled from a sulfidic depth, were bubbled with helium gas for 15 min. $^{15}$N- and $^{13}$C-labeled substrates were added after 5 min of purging in the following experiments: exp1: $^{15}$N-NO$_3^-$ + $^{13}$C-HCO$_3^-$; exp2: $^{15}$N-NO$_2^-$ + $^{14}$N-NH$_4^+$ + $^{13}$C-HCO$_3^-$; and exp3: $^{15}$N-NH$_4^+$ + $^{14}$N-NO$_2^-$ + $^{13}$C-HCO$_3^-$. Concentrations of labeled substrates were 25, 5, and 5 μm for NO$_3^-$, NO$_2^-$, and NH$_4^+$, respectively. At sulfidic depths serum bottles were not bubbled with gas in order to maintain ambient sulfide concentrations, instead, labeled substrates were mixed by stirring. Serum bottles were overflown two times into small glass vials (Exetainers, Labco Limited; 6 or 12 mL) and capped. The caps 2–3 days prior to use were stored in a pre-degassed Duran bottle filled with a helium atmosphere, to reduce oxygen contamination in the incubation experiments[51]. Exetainer incubation experiments were incubated at 12 °C in the dark. After adding a 2 mL helium headspace, Exetainer samples were terminated at 0, 6, 12, 24, and 48 h by the addition of 100 μL of saturated mercury chloride solution. (See the following publications[50–52] for discussions of incubation methods and potential artefacts.) Terminated incubation samples were stored cap down at room temperature. For nanoSIMS analysis a separate 24-h incubation vial was terminated by the addition of a 20% paraformaldehyde solution to a final concentration of 1–2%.

Isotopic ratios of $^{15}$N$^{15}$N and $^{15}$N$^{14}$N dinitrogen gas were measured from the headspace of the incubation experiments using a gas-chromatography isotope-ratio mass spectrometer (GC-IRMS; VG Optima, Manchester, UK). The nitrite production was determined from amended $^{15}$NO$_3^-$ experiments performed by converting labeled nitrite to $^{14}$N$^{15}$N gas[53]. The converted N$_2$ gas was measured by a GC-IRMS (customized TraceGas coupled to a multicollector IsoPrime100, Manchester, UK). Denitrification and anammox N$_2$ production rates were calculated from the linear regression slope as a function of time according to Thamdrup et al.[54]. A $t$-test was used to determine whether rates were significantly different from zero ($p < 0.05$). Detection limits were estimated from the median of the standard error of the slope, multiplied by the $t$-value for $p = 0.05$, the detection limits for anammox, denitrification to N$_2$, and denitrification to NO$_2^-$ from $^{15}$N-labeled experiments were 1.03, 0.13, and 0.80 nM N d$^{-1}$, respectively.

Bulk CO$_2$ fixation rates were determined separately from $^{13}$C-incubation experiments performed in gas tight 4.5 L bottles[5]. To each bottle 4.5 mL of labeled bicarbonate solution (1 g $^{13}$C-HCO$_3^-$ in 50 mL water) was added. Depending on sample depth bottles were incubated at in situ temperatures on-deck in blue shaded incubation boxes (25% surface irradiance) or in the dark. After 24 h, 1–2 L was filtered onto pre-combusted Whatman GFF filters. GFF filters were dried and then treated to remove inorganic carbon by fuming 37% HCl treatment overnight. The isotopic $^{13}$C enrichment was quantified by an element analyzer EA-IRMS (FlashEA 1112 series coupled with an IRMS, Finnigan Delta plus XP, Thermo Scientific). Carbon fixation rates were calculated according to Schunck et al.[5].

**Molecular sampling**. Samples (1–2 L) collected for microbial enumeration by catalyzed reporter deposition FISH (CARD-FISH) were immediately fixed in 20% paraformaldehyde solution to a final concentration of 1–2%. Fixed samples were filtered onboard after 8–12 h at 4 °C onto a 0.2 μm pore-size PC filter. Filtration volumes varied according to depth and location from the coast (i.e., 70–120 mL at offshore stations and 50–70 mL at coastal stations) in order to get adequate cell densities on the filter. Filters for nanoSIMS analysis were collected from $^{13}$C-HCO$_3^-$-labeled incubation experiments (exp1) onto pre-coated gold-palladium 0.2 μm PC filters. For biomass collection and subsequent DNA analysis, larger volumes of sea water (1–2 L) were filtered onto a 0.2 μm PC filters. All filters were stored and transported between −20 and −80 °C.

**DNA extraction, metagenomics, and genome binning**. DNA was extracted from filtered biomass using a DNA/RNA-Allprep kit (Qiagen). Extracted genomic DNA was sequenced with Illumina MiSeq technology and chemistry (Max Planck Institute for Evolutionary Biology, Plön, Germany). Full-length 16S rRNA gene sequences were reconstructed from raw reads using phyloFlash (https://github.com/HRGV/phyloFlash). Adapters and low-quality reads were removed with bbduk (https://sourceforge.net/projects/bbmap/) with a minimum quality value of two and a minimum length of 36, yielding 1 464 909 and 2 143 435 paired-end reads for library preparations from station U1 at depths 30 and 40 m, respectively. Single reads were excluded from the analysis. Single library assemblies were performed using SPAdes 3.90[55] with standard parameters and kmers 21, 33, 55, 77, 99, and 127. Genome binning was performed in Bandage[56] by collecting all contigs linked to the contig that contained the full-length 16S rRNA gene of the SUP05 organism as reconstructed by phyloFlash. The genome completeness for all SUP05 bins was calculated using checkM version 1.07[57] and the gammaproteobacterial marker gene set using the taxonomy workflow. Annotation was performed using prokka[58]. Genes related to nitrate respiration (*nirS*, *narG*, *norB*, and *nosZ*) and carbon fixation (*cbbM*) were visualized on the assembly graph of the SUP05 bin using the Bandage BLAST module with 98% query coverage and 98% identity settings. Read mappings to screen for the presence of the $^U$*T. perditus nosZ* gene

were performed using bbmap with fast = t and minid = 0.95 to only obtain reads reported that come from the same species.

**Clone library and phylogeny**. Universal bacterial primers GM3f (5′-AGAGTTTGATCMTGGC-3′) and GM4r (5′-TACCTTGTTACGACTT-3′) were used to generate full-length 16S rRNA PCR amplicons from DNA samples taken at the redoxcline of sulfidic station U1 (Supplementary Table 2; [59]). Five PCR replicates were done per sample. The PCR conditions were initial denaturation at 95 °C for 5 min, followed by 25 cycles of 95 °C for 1 min, 50 °C for 1 min, 72 °C for 2 min, and a final extension of 72 °C for 10 min. The reactions were run on an Eppendorf Mastercycler gradient PCR machine with a ramp rate of 3 °C s$^{-1}$. The five replicate PCR products were pooled. DNA was visualized by gel electrophoresis and quantified by Nanodrop (Thermo Scientific). The 16S rRNA gene product was purified and ligated into a TOPO TA vector using a ligation kit (Invitrogen). Resulting *Escherichia coli* clones were picked and screened for the vector insert by PCR. Colonies with inserts were regrown in fresh media followed by a plasmid extraction using a plasmid extraction kit (MoBio). The plasmid was amplified in two separate final sequencing reactions using forward and reverse M13 primers (M13f 5′-CCCAGTCACGACGTTGTAAAACG-3′ and M13r 5′-AGCGGATAA-CAATTTCACACAGG3′[60]). The PCR product was purified using Sephadex (G-50 Superfine, Amersham Bioscience) and then sequenced with Sanger sequencing chemistry in-house in Bremen (BigDye sequencing kit, Applied Biosystems).

Raw sequence data were quality controlled and vector ends were trimmed, then forward and reverse amplicons were assembled into near-full-length 16S rRNA contigs using Sequencher 4.6 software (Gene Codes Corporation, Ann Arbor, MI). The 16S rRNA contigs were aligned with the SINA aligner[61], and then imported into SILVARef115 curated 16S rRNA reference database[62] using ARB software[63]. A 16S rRNA tree was calculated using the parsimony and neighbor joining methods using various filters. A CARD-FISH probe (GSO131) was designed in silico using ARB software to target the 16S rRNA gene of SUP05 bacteria recovered from the ETSP region (Supplementary Tables 2 and 3; Fig. 3). The GSO131 probe was tested both in silico and then evaluated on filters collected at station U1 using different formamide contents (10, 20, 30, 40, 50, and 60%) (Supplementary Discussion; Supplementary Fig. 8).

**Fluorescence in situ hybridization**. The CARD-FISH procedure was performed on samples collected on PC filters according to Pernthaler et al.[64]. Briefly, filter pieces were treated with lysozyme (10 g L$^{-1}$) for 45 min at 37 °C to permeabilize the cells for hybridization. The filters were washed in phosphate-buffered saline (PBS) buffer and then Milli-Q water before proceeding to the deactivation of endogenous peroxidases with methanol/hydrogen peroxide (0.15%) treatment for 10 min at room temperature. Samples were washed with Milli-Q before performing the hybridization. Filter pieces were incubated for 3 h at 46 °C in the hybridization buffer containing a 35% formamide concentration. Filters were washed in pre-warmed washing buffer containing NaCl (0.08 M final concentration), 5 mM EDTA (pH 8.0), 20 mM Tris-HCl (pH 7.5), and 0.01% SDS for 15 min at 48 °C then washed again for 10 min in 1× PBS buffer at room temperature. Filter pieces were incubated for 45 min at 46 °C in amplification buffer containing 0.15% H$_2$O$_2$ and 20 μg Oregon Green-labeled tyramide. Filters were washed in 1× PBS and Milli-Q then dried before staining with 4′,6-diamidino-2-phenylindole (DAPI; 1 μg mL$^{-1}$) for 10 min at room temperature. Filter pieces were embedded in a mixture of Citifluor/Vectorshield and DAPI and probe-hybridized signals were counted on an epifluorescence microscope (Zeiss AxioPlan). Up to 1000 DAPI-stained cells from 10 different fields of view were counted. Separate CARD-FISH probes EUB338 and NON338 were used as positive and negative controls, respectively[64].

**NanoSIMS analysis**. Select field of views containing hybridized SUP05 cells were marked using a Laser Microdissection microscope (DM 6000 B, Leica). Isotopic composition of single cells of SUP05 bacteria were analyzed using a NanoSIMS 50L instrument (Cameca). We used the 19 F signal to detect SUP05 cells hybridized with the GSO131 probe to confirm regions of interest. All samples were pre-sputtered with a Cs+ primary ion beam of ~300 pA. After pre-sputtering the instrument was tuned on the target area on a 50 × 50 raster size for a mass resolution over 8000. Secondary ions of $^{12}$C, $^{13}$C, $^{19}$F, $^{12}$C$^{14}$N, $^{12}$C$^{15}$N, $^{31}$P, and $^{32}$S were measured simultaneously on seven electron multiplier detectors with Cs+ primary ion beam of 1.5–2.0 pA. Final analysis and image acquisition were done at 10 × 10 raster size (256 × 256 pixel) and a dwell time of 1 ms per pixel for 40 planes. The data were processed using Look@NanoSIMS software[65]. The field of interest were drift-corrected and accumulated using the software. Cells of interest were interactively defined by hand and classified. For each cell $^{13}$C/$^{12}$C, $^{12}$C$^{15}$N/$^{12}$C$^{14}$N, and $^{32}$S/$^{12}$C+$^{13}$C ratios were calculated. Only cells with Poisson statistics <5% were considered reliable measurements.

NanoSIMS was also performed on an untreated (i.e. no CARD-FISH) filter at station U1 (30 m depth) to determine if the isotopic fraction of $^{13}$C in SUP05 cells was potentially diluted by the CARD-FISH protocol[66]. Based on CARD-FISH enumeration, we assumed that at least half the cells measured by nanoSIMS on the non-CARD-FISH filter were SUP05 bacteria. In this case, no difference in the average isotopic composition was found between the treated (0.19 ± 0.02 fmol C per

cell per day) and untreated ($0.20 \pm 0.05$ fmol C per cell per day; $n = 46$ cells) samples. Sulfur ionizes much better than carbon; furthermore, the ionization of elements is matrix-dependent. Nevertheless, the matrices across all the cell samples were similar, and certainly across the cells identified as belonging to the SUP05 clade.

**Single-cell calculations**. The cell size, determined from Look@NanoSIMS software, was used to estimate the cell biovolume ($V$). We have chosen the more conservative calculation for carbon content per cell volume for bacteria[32]. SUP05 cell biovolume ($V$) was calculated according to the volume of a cylinder with two half-spheres as ends (e.g., a rounded rod[67]).

$$V = \pi B^2 (0.125A - 0.0833B) \qquad (4)$$

where $A$ is the length of the rod and $B$ is the diameter of the rod.

We estimated the cell carbon content according to the allometric relationship between the cell biovolume ($V$) and carbon content[32]:

$$\text{fg C per cell} = 135 V^{0.438} \qquad (5)$$

This equation takes into account the higher carbon content of smaller cells (i.e., because of the minimum set of molecules a cell has, small cells tend to have larger carbon contents per volume than larger cells[32]).

For each cell the fraction C fixed per unit time ($f$) can be determined from the $^{13}\text{C}/^{12}\text{C}$ ratios obtained for individual cells, the known labeling percentage, and the incubation time.

$$f = \text{fraction C fixed per day}$$
$$= \left( {}^{13}\text{C}/{}^{12}\text{C}_{\text{cell}} - {}^{13}\text{C}/{}^{12}\text{C}_{\text{background}} \right) \left( {}^{13}\text{C}/{}^{12}\text{C}_{\text{DIC}} \right)^{-1} t^{-1} \qquad (6)$$

where $^{13}\text{C}/^{12}\text{C}_{\text{cell}}$ is the $^{13}\text{C}/^{12}\text{C}$ ratio on cells measured by nanoSIMS, $^{13}\text{C}/^{12}\text{C}_{\text{background}}$ is the background natural abundance of unlabeled cell material, and $^{13}\text{C}/^{12}\text{C}_{\text{DIC}}$ is the fraction of dissolved inorganic carbon labeled with $^{13}\text{C}$, and $t$ is incubation time in days.

The single-cell carbon assimilation rate is calculated by multiplying Eq. 6 by the per cell C content.

$$\text{Cell specific rate of C fixation} = f \times \text{fmol C per cell} \qquad (7)$$

SUP05 bacteria-associated $CO_2$ fixation was calculated using the single-cell $CO_2$ fixation rate (fmol C per cell per day) and the SUP05 cell densities (cells per L). The percent contribution of SUP05 to bulk carbon fixation was calculated from the SUP05 $CO_2$ fixation rate divided by the bulk $CO_2$ fixation rate.

**Remote sensing imagery**. Chlorophyll remote sensing imagery was downloaded from the Moderate Resolution Imaging Spectroradiometer and Visible Infrared Imaging Radiometer Suite databases using NASA Ocean Color[68]. Level 2 and 3 data were processed using SeaDAS software version 7.3.1 [https://seadas.gsfc.nasa.gov/]. Sea-surface satellite altimetry images were downloaded from the Colorado Center for Astrodynamics Research [https://eddy.colorado.edu/ccar/ssh/nrt_global_grid_viewer].

**Data availability**. The SUP05 Whole Genome Shotgun project has been deposited at DDBJ/ENA/GenBank under the accession PNQY00000000. The version described in this paper is version PNQY01000000 [https://www.ncbi.nlm.nih.gov/nuccore/1334810922]. The gammaproteobacterial sulfur-oxidizing 16S rRNA genes were submitted to the NCBI database under the accession number MG518493-MG518517. Water column nutrients and physical data are available at Pangaea [https://doi.pangaea.de/10.1594/PANGAEA.860727]; while station sulfur chemistry, SUP05 cell densities, and rate process measurements have been submitted to Pangea [https://doi.pangaea.de/10.1594/PANGAEA.876062].

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

## Acknowledgements

We are grateful to the Peruvian authorities, T. Kanzow, the captain and the crew of the RV Meteor. We thank P. Lam and G. Klockgether for extensive onboard experimental and analytical support, and C. Schelten for administrative support. M. Dengler graciously provided pre-publication eddy diffusion coefficients, and the MPI Plön assisted with sequencing. Four thoughtful, constructive reviews greatly benefited the manuscript. We acknowledge the generous access to data afforded by the NASA Ocean Biology Processing Group and the Colorado Center for Astrodynamics Research. This work was supported by the Max Planck Society and the German National Science Foundation (DFG) Sonderforschungsbereich (SFB754) GEOMAR, Kiel. C.M.C. was supported by a Natural Sciences and Engineering Research Council of Canada (NSERC) scholarship and C.R.L. by EU/ H2020 grant #704272, NITROX.

## Author contributions

C.M.C., G.L., B.F., H.R.G.-V., S.T., and M.M.M.K. designed the study; C.M.C., H.R.G.-V., P.F.H., S.L., N.J.S., T.K., S.T., and H.S. performed experiments; C.M.C., G.L., T.G.F., H.R.G.-V., P.F.H., S.L., N.J.S., T.K., S.T., H.S., C.R.L., and R.A.S. analyzed data; C.M.C., T.G.F., G.L., and M.M.M.K. wrote the manuscript with input from all co-authors.

## Additional information

**Competing interests:** The authors declare no competing interests.

