## [Peer Review File · Nature Communications]

Reviewers' comments:

Reviewer #1 (Remarks to the Author):

Overall, I give the manuscript titled "Oxygen minimum zone cryptic sulfur cycling sustained by offshore transport of key sulfur oxidizing bacteria" a high rating. I do have one major issue that needs to be addressed in detail and some more general comments that need to be addressed prior to publication in Nature Communications.

Major issue:

The authors list carbon content/cell (Table 1, Line 5) as $6.42-8.46 \times 10^{-3}$ pmols, which is also fmoles. This is much too high for marine bacteria. Carbon concentrations in marine bacteria range from 10-30 fgrams C/cell, which when converted to fmols is 0.83 to 2.5 fmols. In the methods the authors state that they used the equation ($\text{fg C cell}^{-1} = 133.754 \times \text{VE}^{0.438}$). I attempted reproduced the calculations and it looks like they did not convert grams carbon/cell (dry weight) to moles carbon/cell in Table 1, line 5. If that is correct, it has significantly altered the findings, with implications for SUP05 carbon fixation rates, contributions to total carbon fixation, and subsequent nitrate and sulfide utilization rates, which are based on a growth factor derived from carbon fixation. If pmols is a typo and it should say pgrams, then the authors need to provide more details on the calculations, so that they can be reproduced. More on the calculations would be useful, in general. As is, it's hard to tell exactly what was used in the final calculations.

General comments:

First, while the manuscript presents some novel findings, the context, as reflected in the title and elsewhere, is a bit confusing. It is unclear to me if the authors agree or disagree with the concept of the cryptic sulfur cycle. In places the manuscript suggests clarity by providing a physical mechanism that helps explain the cryptic sulfur cycle. In other places the manuscript suggests that the concept of the cryptic sulfur cycle is not correct, or at least not accurate. I assume it's the later, mostly because an entire paragraph is devoted to the lack of evidence for sulfur reduction, which is the other half of the cryptic sulfur cycle. I suggest that the authors clarify their position in the title and elsewhere. Does the manuscript present an alternative explanation or does it elucidate processes underlying an existing one? Second, the authors have some very nice data, but also make a lot of assumptions in reaching their conclusions. Much of the data (such as the metagenomics data and "growth factor" estimates) come from a sulfidic sample on the shelf (U1). These data are the basis for some assumptions about SUP05 in other locations. Third, there are several places where the authors highlight the significance of their findings by suggesting that something is unknown, when in fact it is well known. For example, an abundant sulfur-oxidizing, nitrate reducing, chemolithoautotroph is by definition important in the carbon, nitrogen and sulfur cycles. This is well known. Stating that they "may" be important in these cycles is not accurate. I think the data are strong enough to remove the straw man statements. Finally, a Candidatus genus and species name should not be given to partial or even complete genome sequences assembled from metagenomes. They are chimeric and uncultured. Please see more specific comments below for some examples of these points.

Specific comments:

Line 33: Genus species names, even Candidatus, should not be assigned to a group of organisms identified by metagenomics. Complete or nearly complete genomes assembled from metagenomic data are chimeric in nature, even if the 16S rRNA gene is 100% identical for all cloned pieces of DNA. Naming is typically for isolates, though it has been done in the case of endosymbionts, which can't be cultured independently of their host. I think it gives the wrong impression to name a group, as indicate in Figure S1. This creates confusion with respect to future studies.

Lines 34-37: This sentence should be reworded. It becomes unclear at line 36, at "underpins". Maybe just by starting a new sentence "This underpins . . ."

Lines 47-50: "May" or "do" substantially contribute. The references suggest that this is known. What about Hawley, 2014 (reference 12), which estimates carbon fixation rates for SUP05.

Line 51: Need to add "with" to "Genes associated the SUP05 . . ."

Line 56: Why "so-called". In places the authors give the impression that they verify the cryptic sulfur cycle and in other places they seem to oppose the concept. It's a little confusing. Does this study support, clarify, or debunk the "cryptic sulfur cycle"? Is this an alternative hypothesis?

Line 65: Published data are stronger than "appear". It might be better to say "It's clear that SUP05 have important roles in . . ., but several important questions remain".

Lines 70-73: This is another place where the authors seem to disagree with the cryptic sulfur cycle concept, by suggesting that existing probes are not specific enough. Much of the data for the cryptic sulfur cycle are based on functional genes. Some of which are not present in Arctic96BD-19 genomes. Please clarify.

Lines 73-74: This can be said about every major group of marine bacteria in every major marine system.

Lines 83-86: This to me really says that the authors do not agree with the concept of the cryptic sulfur cycle, because it "may simply reflect the advection of sulfur". I guess my point/question is: Doesn't this critical sentence contradict the title of the paper?

Line 132: In "traveled already further westward", I would remove the phrase "already further".

Figure 2: Some of the highest concentrations of SUP05 are in areas on the shelf that are 20-40 km from the coast and deeper than the elemental sulfur and denitrification maxima. These areas have no detectable sulfide, relatively little elemental sulfur, little or no evidence of denitrification, but some evidence of SUP05 carbon fixation. Does this provide evidence of a cryptic sulfur cycle? Meaning lots of SUP05, some carbon fixation, but no detectable sulfide. Where these SUP05 populations evaluated by 16S rRNA gene analyses? Are they the same as the SUP05 in the metagenome?

Lines 177-179: What about other SUP05, not Arctic96BD-19 or Peru upwelling SUP05? Are other species or strains potentially present that could have different metabolic capabilities. It would be helpful to know how good the probe is at discriminating all SUP05 from Arctic96BD-19. The data present suggest that Peru upwelling SUP05 are the dominant ecotype at U1 (30m), which is also the denitrification maxima and has the highest elemental sulfur concentrations. My comments above suggest that other SUP05 may be present. How do the authors know this group of SUP05 are the ones offshore? Also, the Peru upwelling SUP05 appear to have the same 16S rRNA gene sequences (Figure S1). So does the fact that the probe hits 10 SUP05 sequences really mean that it hits multiple copies of the same sequence?

Line 212: Are they correlated or not correlated? Are there any statistics? This is misleading. Especially since the next sentence (topic sentence of a paragraph) starts with "Despite such correlation". This moves it from appears to is correlated. I don't think this is accurate.

Line 227: How does this compare with other estimates of carbon fixation in SUP05 (Hawley et al., 2014).

Line 258-259: Supports or does not support the cryptic sulfur cycle idea?

Lines 260-262: Since SUP05 are well known nitrate reducing and sulfur oxidizing chemolithoautotrophs, I think the answer is clearly yes. It would be better to remove some of the

straw man sentences and just state that SUP05 has important roles in marine carbon, nitrogen and sulfur cycling (ref).

Line 265: This is a big assumption because it assumes that coastal SUP05 and transported SUP05 are the same cells (Peru upwelling SUP05) with the same genetic potential and physiological responses to those measured at U1. How do published estimates for carbon fixation and nitrate utilization of SUP05 compare with the result reported in this manuscript (Hawley et al, 2016; Shah et al, 2016)?

Lines 283-285: Against cryptic sulfur cycle?

Line 290: All metagenomics data are from U1. Where 16S rRNA analyses conducted elsewhere and compared to those in the assembled genome? Meaning, how do the authors know these SUP05 are the same ones present elsewhere? If the new SUP05 probe specific to Peru upwelling SUP05 at U1 was used for single cell analyses at offshore sites, then it would be critical to know if these were the only SUP05 present at all other stations.

Line 303: Why oxygen, I don't see evidence of aerobic respiration in the figure or in the text.

Line 305: The only complete genome sequence, *T. autotrophicus* is not included. Should probably reference either Shah and Morris 2015; Shah et al., 2016.

Line 308: It's *T. autotrophicus* not *T. autotrophica*

Line 311: See earlier comment regarding naming.

Line 314: This paragraph seems to be the argument against the cryptic sulfur cycle.

Lines 350-353: This is where a lot of assumptions come together to make a statement that is somewhat misleading. This sentence suggests that a specific species of SUP05 capable of complete denitrification is the only group that is transported and that is active.

Line 536-540: The growth factor was calculated at U1, 30m, where there is peak denitrification and peak denitrifying SUP05. This factor was then used to estimate values at other stations. This is where most of the assumptions come together. Another assumption is that sulfide, or elemental sulfur, is fully oxidized to sulfate. Do we know this? How do we know that the SUP05 populations are the same, and that the physiology of SUP05 cells are the same?

Reviewer #2 (Remarks to the Author):

This is a really nice paper and I suggest publication in its current form. Understanding the biogeochemical capacity of SUP05 is really important and this study does a fantastic job at (honestly) taking that task head-on.

There is a great accounting and involvement of documenting water mass movement and eddy development / propagation - this ends up being central to the story and is under-appreciated in previous work. Although the study, at its heart, is looking to understand the coupling between N and S, the calculations on environmental growth factors provide a nice check on the overall stoichiometry, and the quantified contributions to the carbon cycle are a nice addition. In the end, the necessary reduced sulfur compounds are reportedly being transported via mesoscale eddies and not local "cryptic" sulfate reduction. This is super cool, and the authors should be clear that this actually goes against the logic reported in Canfield et al., where they call on local sulfate reduction and immediate sulfide oxidation. That is essentially the only criticism - the abstract is vague with respect to how these results impact earlier proposals. Finally, an additional sentence would be nice outlining the potential environmental role for thiosulfate, given that the genetic

complement is present in SUP05. I presume that concentrations were below detection?

Reviewer #3 (Remarks to the Author):

This manuscript describes an interesting set of field measurements of N, C, and S cycle processes in the ETSP. The manuscript is generally well-written, though some of the points were hard to tease out of the text. For example, I am unsure how to interpret the elemental sulfur concentrations given here: they are reported as if they're measurements of dissolved S, but in fact must include substantial particulate S₀. Similarly, I didn't learn until I got to the methods section that the rate measurements reported are really potential rates under anoxic conditions rather than estimates of what's going on in the water column. This is an important distinction and really shouldn't be buried in the methods section. On the whole, the findings are interesting, but the role of this cryptic S cycle in the water column isn't clear to me given the experimental approach used.

A number of more specific comments and suggestions follow.

Figure 1A: The black arrows don't show up very well against the dark blue. I suggest either adding a light border or using a different color altogether.

Figure 1c: It would be helpful to have contours or some other indication of the depth of the water mass shown here. Even with that, I'm not sure that this panel adds anything meaningful to the paper.

p. 6, first two paragraphs: The stations referenced should be clearly identified in Figure 1.

p. 6, paragraph 3: Elemental sulfur is highly insoluble in water and the μM S₀ concentrations reported are orders of magnitude higher than the equilibrium solubility. The analytical methods used to measure elemental sulfur concentrations (p. 16) will capture both dissolved and particulate S₀, and the bulk of the reported concentrations must be accounted for by solid phase S₀. Given the apparent importance of elemental sulfur in this story, I'm surprised at the lack of any effort to get at the solid/dissolved partitioning of S₀, which might affect the movement of S₀ via mixing (e.g., p. 6, final paragraph) as well as the accessibility of S₀ to organisms.

p. 6, final paragraph: An estimate of the potential rate of supply of S₀ to the upper chemocline would provide important context here. Specifically, can eddy diffusion provide enough S₀ to sustain meaningful rates of coupled denitrification?

Figure 2: I suggest tweaking the color scales so that the highest concentrations are represented by the deepest red; it's visually confusing to have the scale transition to lighter colors for the highest concentrations.

Figure 2: Panel d shows that the bulk of the population of SUP05 is out on the outer shelf, while Panels e-f shows little or no activity associated with SUP05 out there. Why is the deeper population so inactive?

p. 7: This sort of calculation is exactly what I'd like to see for the postulated S₀ flux to the top of the chemocline.

p. 8, second paragraph: I don't see Stn L2 in Fig 2. Since this is an important reference station, it really needs to be shown and clearly identified

p. 9, first paragraph: this is an important point, but it's not at all obvious from Figure 2. Stn L2 really needs to be added to that figure to make this point visually.

Table 1: I suggest explicitly showing the relative contribution of SUP05 to carbon fixation (about 65% if I'm reading the table correctly).

Table 1: What is the value of depth-integrated SUP05 carbon fixation that's given in parentheses (8.6 ± 0.7)?

Table 1: For the final category, why not provide an estimate of the SUP05 contribution to the bulk rates. A quibble: these are areal rates, not turnover estimates.

Figure 3: For clarity, please be consistent in ordering of depths from left to right (L1 is deep-shallow while U1 is shallow-deep).

p. 11, 2nd sentence: I understand that the rates at L1 and U1 are not significantly different, but isn't it interesting and worth noting that the specific rates are not just comparable, but actually tend to be higher at the offshore station?

p. 12, paragraph 2, sentence 3: What evidence is there that denitrification is primarily due to SUP05? Earlier, the authors estimate that up to 70% of the downward nitrate flux could be consumed by sulfide oxidation, presumably by SUP05 (p. 7, last paragraph). Since 70% is an upper limit, treating all of the denitrification as due to SUP05 will bias the estimates of this organism's impact on the N cycle.

p. 13, first paragraph: What is the physical distribution of S in SUP05 cells? The nanoSIMS analysis should provide some insight into whether the S is stored in granules (e.g., S₀ granules), or in some other form. It'd also be worth mentioning any association with either O or N, which could also provide clues to the physical form of the stored S.

p. 17, second paragraph: Sparging with He will remove any O₂ present, which will promote anaerobic processes. This really needs to be brought up well before the methods section, since the measured rates of denitrification and anammox are going to be elevated relative to what's going on in the water column. The rates reported are potential, not in situ rates.

=====
=====End of Review=====

Reviewer #4 (Remarks to the Author):

This is an exciting and well designed study that extends our understanding of the ecology and physiology of the globally important sulfur-oxidizing bacterial SUP05 group. Set in the highly dynamic Peru upwelling region, this work demonstrates changes in the physiology and activity of the SUP05 through the influence of physical and chemical oceanographic processes while also shaping the cycling of C, N and S within the oxygen minimum zone. The manuscript is coherently written, quantitative, and incorporates compatible, state-of-the-art methods. I have only a few questions, minor comments and suggestions for further strengthening this manuscript.

Line 230: If the SUP05 represented ~65% of the total dark CO₂ fixation, did you see evidence of other cells in addition to those targeted by the SUP05 probe that showed active ¹³C-bicarb incorporation by nanoSIMS?

Similarly,

Lines 255-256: Can you clarify this comment about station L2? does this mean there are other organisms contributing to dark C fixation in addition to SUP05, or is there very low rates of C fixation overall?

Line 300: From the nanoSIMS data was there any visible difference in the S distribution associated with the SUP05 cells? Any extracellular precipitation observed, or just intracellular S/C ratios? Again, were any other bacteria analyzed from these samples and do you have data on their S/C ratio as a point of comparison (i.e. is this elevated S/C ratio unique to SUP05 or is there higher S/C for other cells in the same sample)?

Line 305-306: is this statement based on the genome prediction or your activity based measurements? If associated with the genome data, I suggest rewording the sentence indicating this is a predicted genomic capability rather than stating that they are capable of complete denitrification coupled with sulfide and elemental sulfur oxidation. From the data presented in this study, none of the specific physiology experiments directly indicates their ability to use elemental sulfur.

Line 310: do you have an ANI percentage you can provide for the similarity between *T. autotrophica* and peru upwelling SUP05 genome bin?

Line 438: Need to provide information about which samples were prepared for DNA extraction/metagenomics. How was the sample concentrated and what was the volume?

Lines 478-480: Was the specificity of the newly designed GSO131 probe tested aside from in silico predictions? If so, add a comment here. Also given that this is a new probe, if you have any information about optimization that would also be good to include (beyond the footnote in Table S2).

Line 500: The CARD-FISH procedure includes a methanol /H₂O₂ treatment and possibly a ethanol dehydration series which can leach elemental sulfur inclusions from cells- do you have any sense of how this may have impacted your S/C ratio data and does this have implications for the form of sulfur observed in the cells if it wasn't removed during the dehydration step?

Line 502: Did you use the Oregon green 19F signal to confirm the hybridization signal by nanoSIMS? If so, please include that information here and cite a reference.

Line 507: What was the pA value for the Cs⁺ ion beam during the analysis?

Overall, the nanoSIMS experiments provided important insights into the physiology of these organisms across the different stations. Additional control experiments including tests of potential shifts in cellular ¹³C/¹²C ratio from exogenous carbon introduced during CARD-FISH hybridization were also useful.

Some information about cellular ¹⁵N data would be worth commenting on in the text given that this was one of the isotope ratios measured- even in the absence of a trend, it would be helpful to know what that data looked like. Also some discussion on variation in secondary ion production for sulfur vs carbon is warranted in the context of interpretation of S/C cell ratios.

Figure 2. Much of the data is associated with station L2, but this station is not included in the figure.

Authors' Response to Reviewer Comments:

Oxygen minimum zone 'cryptic sulfur cycling' sustained by offshore transport of key sulfur oxidizing bacteria

Cameron M. Callbeck et al.

Authors' Response to Reviewer Comments:

We would like to thank the reviewers for their thorough and constructive comments. We have addressed their comments and made changes in the text where necessary. Below we provide a point-by-point response to the reviewers' comments and questions. In particular, we have addressed the issue of what our result imply for the cryptic sulfur cycle; we have more carefully delineated the identity of SUP05 bacteria offshore in the Peru Upwelling; and we have revisited the calculations on C fixation and denitrification. In the course of these revisions, the abstract and final section of the manuscript have been completely rewritten, we have moved a new version of the phylogenetic tree (formerly Figure S1; Phylogenetic diversity of GSO 16S rRNA genes) into the main text; and we have redone Table 1, which includes the single-cell C-uptake calculations.

We believe that the revisions have improved the manuscript, and we thank the reviewers for their important role in improving the manuscript.

Reviewer #1 (Remarks to the Author):

Overall, I give the manuscript titled "Oxygen minimum zone cryptic sulfur cycling sustained by offshore transport of key sulfur oxidizing bacteria" a high rating. I do have one major issue that needs to be addressed in detail and some more general comments that need to be addressed prior to publication in Nature Communications.

Major issue:

The authors list carbon content/cell (Table 1, Line 5) as $6.42-8.46 \times 10^{-3}$ pmols, which is also fmoles. This is much too high for marine bacteria. Carbon concentrations in marine bacteria range from 10-30 fgrams C/cell, which when converted to fmols is 0.83 to 2.5 fmols. In the methods the authors state that they used the equation ($\text{fg C cell}^{-1} = 133.754 \times \text{VE}0.438$). I attempted reproduced the calculations and it looks like they did not convert grams carbon/cell (dry weight) to moles carbon/cell in Table 1, line 5. If that is correct, it has significantly altered the findings, with implications for SUP05 carbon fixation rates, contributions to total carbon fixation, and subsequent nitrate and sulfide utilization rates, which are based on a growth factor derived from carbon fixation. If pmols is a typo and it should say pgrams, then the authors need to provide more details on the calculations, so that they can be reproduced. More on the calculations would be useful, in general.

As is, it's hard to tell exactly what was used in the final calculations.

We have double checked our calculations and are certain that our calculations are correct..

We have chosen the more conservative calculation for carbon content per cell volume for bacteria provided by (Romanova and Sazhin, 2010). Given that the SUP05 cells were coccoid we calculated the cell biovolume according to the volume of a sphere

$$V=4/3\pi r^3 \quad [\text{Eq. 1}]$$

The SUP05 cells at station U1 averaged 0.81 μm in diameter ($r = 0.41 \mu\text{m}$). We therefore obtain a biovolume of $0.28 \mu\text{m}^3$. We estimate the cell carbon content according to the allometric relationship between the cell biovolume (V) and carbon content proposed by Romanova and Sazhin, 2010:

$$\text{fg C cell}^{-1} = 133.754 \times V^{0.438} \quad [\text{Eq. 2}]$$

This equation takes into account the higher carbon content of smaller cells (i.e. because of the minimum set of molecules a cell has, small cells tend to have larger carbon contents per volume than larger cells)(Romanova and Sazhin, 2010). If we input the cell biovolume (V) into [eq. 2] we obtain $76 \text{ fg C cell}^{-1}$ for station U1 (and $101 \text{ fg C cell}^{-1}$ for station L1). These estimated carbon content values for SUP05 bacteria are within the range reported for marine bacteria ($7\text{-}260 \text{ fg C cell}^{-1}$) in other eutrophic coastal environments (Fukuda et al., 1998 and references therein).

The value of 10-30 fg carbon per bacterial cell suggested by the reviewer is appropriate for an average sized marine bacteria. Such a value comes from Lee and Fuhrman (1987), who reported an average of 20 fg cell^{-1} for cells with a biovolume of $\sim 0.05 \mu\text{m}^3$ ($380 \text{ fgC}/\mu\text{m}^3$). Carbon content, of course, depends on the cell size, and has been shown to vary over several orders of magnitude depending on cell size (with an average of $148 \text{ fgC}/\mu\text{m}^3$, Gundersen et al 2002). The average cell volume of the SUP05 reported here are almost 20 times larger ($0.8\text{-}1.0 \mu\text{m}^3$, Table 1) than those reported in Lee and Fuhrman (1987) with a expectedly greater carbon content. We then divide $76 \text{ fg C cell}^{-1}$ by the molecular weight of carbon (12.01 g mol^{-1}) to obtain $6.36 \text{ fmol C cell}^{-1}$. To calculate a single-cell carbon assimilation rate (see Eq. 3) we use the estimated SUP05 carbon content (above), the SUP05 ^{13}C enrichment (determined from nanoSIMS analysis), the labelling percent, divided by the incubation time.

$$((^{13}\text{C excess} \times \text{fmol C cell}^{-1}) / \text{labelling percent}) / (1 \times \text{incubation time}) \quad [\text{eq 3.}]$$

If we input the SUP05 carbon content for station U1 along with the other measured values (average: ^{13}C SUP05 excess = 0.0024; labelling percent = 0.067; incubation period = 0.96 days) into Eq. 3 we arrive at a single-cell SUP05 assimilation rate of $0.24 \text{ fmol C cell}^{-1} \text{ d}^{-1}$.

We have provided a more detailed calculation in the material and methods section of the revised manuscript (lines 571-596).

General comments:

First, while the manuscript presents some novel findings, the context, as reflected in the title and elsewhere, is a bit confusing. It is unclear to me if the authors agree or disagree with the concept of the cryptic sulfur cycle. In places the manuscript suggests clarity by providing a physical mechanism that

helps explain the cryptic sulfur cycle. In other places the manuscript suggests that the concept of the cryptic sulfur cycle is not correct, or at least not accurate. I assume it's the later, mostly because an entire paragraph is devoted to the lack of evidence for sulfur reduction, which is the other half of the cryptic sulfur cycle. I suggest that the authors clarify their position in the title and elsewhere. Does the manuscript present an alternative explanation or does it elucidate processes underlying an existing one?

*Here and elsewhere, the reviewer raises the issue of whether we “agree or disagree with the concept of the cryptic sulfur cycle”. The main aim of this manuscript is to understand the role of SUP05 bacteria, such as *T. perditus*, in coupling S and N cycling in offshore OMZ waters, not to support or debunk the idea of a “cryptic sulfur cycle”. Our data and observations suggest that while an OMZ cryptic sulfur cycling may exist, it is only minimally supported by local sulfate reduction. Rather, offshore transport of elemental S and SUP05 can account for observed cryptic S cycling in offshore waters. . We have made these statements more forceful, both in the abstract and in the final section, the latter of which has also been retitled as: “Implications of cross shelf-transport for ‘cryptic sulfur cycling’”. We underscore is that SUP05, a key organism involved in the coupled S-N cycles, is indeed active in offshore OMZ waters. They can maintain their activity, because in spite of being swept offshore they can use elemental sulfur that has been transported along with them as a substrate. Our results do not disprove other scenarios, i.e. concurrent sulfate reduction and sulfide oxidation, but do provide another straightforward explanation for the presence and the activity of these organisms offshore.*

We thank the reviewers for forcing us to more closely think about this issue.

Second, the authors have some very nice data, but also make a lot of assumptions in reaching their conclusions. Much of the data (such as the metagenomics data and “growth factor” estimates) come from a sulfidic sample on the shelf (U1). These data are the basis for some assumptions about SUP05 in other locations.

There are two major assumptions that the reviewer alludes to here. This first refers to the identity and similarity of the SUP05 populations at the shelf and offshore sites. Are the SUP05 bacteria active at the offshore site L1 the same as those at the near-shore sulfidic site U1? Yes, they are. We answer this comprehensively in our response to the reviewer comment for Lines 177-179 below.

The second refers to assumptions regarding the calculation of environmental growth factors. Here, we agree with the reviewer. Whereas the estimates of SUP05 specific carbon fixation are very robust, we agree with the reviewer that the further calculations concerning environmental growth factors were difficult to follow and they involved very large uncertainties. In going through them once more we also uncovered a mistake in one of the growth factor calculations. Therefore, we have moved back one step and have used growth factors obtained from sulfide oxidizing enrichment cultures. We are careful to state the assumptions built into these estimates. We realize that there are limitations, nevertheless, the results are informative and help to convey the importance of SUP05 bacteria for OMZ sulfide oxidation and denitrification.

Concerning the limitations and assumptions we write at lines 290-301:

“There are considerable uncertainties associated with these assumptions, because SUP05 may perform

incomplete sulfide oxidation (as per equations 1 and 2), nitrate reduction may stop at nitrite, the environmental growth yields may be substantially lower than those estimated from pure culture organisms, and SUP05 may have the capacity to perform aerobic sulfide oxidation (see below). Nonetheless, the calculations of potential SUP05 mediated nitrate reduction are informative. The rates of SUP05 mediated denitrification as upscaled from the single-cell C fixation rates fall within the range of rates of nitrate reduction to N₂ or to nitrite determined from bulk 15N experiments at all three stations (Table 1). Thus, the SUP05 single-cell CO₂ uptake data demonstrate that SUP05 was capable of supporting rates of denitrification and sulfur oxidation observed in offshore ETSP waters, especially in water masses that had recently originated from sulfur rich near-shore regions.””

We have also reconstructed Table 1, which we believe now is easier to follow.

Third, there are several places where the authors highlight the significance of their findings by suggesting that something is unknown, when in fact it is well known. For example, an abundant sulfur-oxidizing, nitrate reducing, chemolithoautotroph is by definition important in the carbon, nitrogen and sulfur cycles. This is well known. Stating that they “may” be important in these cycles is not accurate. I think the data are strong enough to remove the straw man statements.

We have revised statements in the text where necessary. See, for example, our response to the reviewer comment for line 65 below.

Finally, a Candidatus genus and species name should not be given to partial or even complete genome sequences assembled from metagenomes. They are chimeric and uncultured. Please see more specific comments below for some examples of these points.

See comment immediately below.

Specific comments:

Line 33: Genus species names, even Candidatus, should not be assigned to a group of organisms identified by metagenomics. Complete or nearly complete genomes assembled from metagenomic data are chimeric in nature, even if the 16S rRNA gene is 100% identical for all cloned pieces of DNA. Naming is typically for isolates, though it has been done in the case of endosymbionts, which can't be cultured independently of their host. I think it gives the wrong impression to name a group, as indicate in Figure S1. This creates confusion with respect to future studies.

We respectfully disagree. We believe that by naming the SUP05 organism identified as being active in the Peru Upwelling OMZ will reduce the confusion surrounding the nomenclature SUP05/GSO/Arctic. We base our decision in large part on a recently published paper by Konstantinidis et al. (ISMEJ 2017) that proposes rules for the classification of well characterized metagenomic bins. In our manuscript we fulfill the suggested requirements:

(1) >80% completeness (here 95%) with

(2) ecological data (in the paper)

Additionally we fulfill part of the following optional requirements:

(3) almost complete 16s rRNA gene sequence (>1400 bp; in the tree)

(4) experimental data confirming bioinformatics predictions (in the paper)

(5) a picture of the organism by FISH (in the Supplements, Fig S6cd)

In the text we replace the "Candidatus" with the prefix "U" as suggested by Kostantinidis et al., ISME J, 2017.

The current 'confusion' derives from previous studies that have neglected already published sequences and clade names, as well as non-thorough phylogenetic analyses. We have here assembled all available sequence data and have constructed a consensus tree. We are convinced our analysis will actually help to clarify phylogeny and summarizes the taxonomy of these clades.

Lines 34-37: This sentence should be reworded. It becomes unclear at line 36, at "underpins". Maybe just by starting a new sentence "This underpins . . ."

Thank you for catching this. We have rewritten the abstract.

Lines 47-50: "May" or "do" substantially contribute. The references suggest that this is known. What about Hawley, 2014 (reference 12), which estimates carbon fixation rates for SUP05.

We have changed this sentence to more clearly reflect that several studies demonstrate that SUP05 has the potential to do denitrification, fix carbon and release N₂O based on genomic inferences, nevertheless, their actual impact has not been directly quantified as done in our study.

We now write (lines 49-51):

"The extent to which SUP05 organisms are active and directly contribute to sulfide oxidation, dark carbon fixation and fixed nitrogen loss, has not been directly quantified thus far. "

Line 51: Need to add "with" to "Genes associated the SUP05 . . ."

Added.

Line 56: Why "so-called". In places the authors give the impression that they verify the cryptic sulfur cycle and in other places they seem to oppose the concept. It's a little confusing. Does this study support, clarify, or debunk the "cryptic sulfur cycle"? Is this an alternative hypothesis?

The phrase cryptic element cycle implies a local production and consumption of a compound, e.g. hydrogen sulfide, which is kept at low to non-detectable levels. This is actually a basic tenet of biogeochemistry, and has been for decades. It is nonetheless, , a useful term for forcing us (and others) to more closely examine local and tightly controlled biogeochemical processes, especially where we initially do not suspect them. That being said, a cycle is no longer cryptic if one can "see" or distinctly measure one of the key intermediate reactant, e.g. elemental S. Nor is it by definition cryptic if the substrate is not entirely locally produced but imported from elsewhere (in this case from the shelf).

Therefore, we continue to employ “so-called” and have even moved it into the abstract, and also put ‘cryptic sulfur cycle’ in inverted commas to emphasize that sulfur cycling offshore in the OMZ does take place, but it is not necessarily cryptic.

Line 65: Published data are stronger than "appear". It might be better to say "It's clear that SUP05 have important roles in . . . , but several important questions remain".

We have changed the sentence to read (Lines 68-70):

“The SUP05 clade link nitrogen and sulfur cycling in OMZ water, nevertheless, several important questions regarding the distribution, metabolic capabilities and actual activities of SUP05 persist.”

Lines 70-73: This is another place where the authors seem to disagree with the cryptic sulfur cycle concept, by suggesting that existing probes are not specific enough. Much of the data for the cryptic sulfur cycle are based on functional genes. Some of which are not present in Arctic96BD-19 genomes. Please clarify.

We are not directly addressing the concept of a cryptic sulfur cycle here. Rather we are addressing the very broad coverage of the GSO/SUP05/Arctic Assemblage GSO477 FISH probe.

We made this statement more pointed (Lines 74-77):

An accurate census of SUP05 cell abundances in OMZ waters is absent, in part because the fluorescent in situ hybridization (FISH) probe (GSO477) previously employed to identify SUP05 bacteriatargets other sulfide-oxidizing bacteria, for instance the heterotrophic sulfide oxidizing Arctic96BD-19 clade (22). “

Lines 73-74: This can be said about every major group of marine bacteria in every major marine system.

Changed this sentence to read (Lines 78-89):

“Moreover, the capacity of marine OMZ SUP05 bacteria to perform partial or full denitrification, for instance in the ETSP or other marine upwelling ecosystems has not been determined.”

Lines 83-86: This to me really says that the authors do not agree with the concept of the cryptic sulfur cycle, because it "may simply reflect the advection of sulfur". I guess my point/question is: Doesn't this critical sentence contradict the title of the paper?

Our point here is that advection of water masses from sulfur rich onshore regions provides a quantitatively important explanation for the presence and activity of these communities offshore.

The sentence does not contradict the title, especially now that we have put ‘cryptic sulfur cycle’ into inverted commas. This is simply a hypothesis. We realized that we had the chance to directly test this hypothesis, because we were present when an eddy drove cross-shelf transport, and we were able to compare SUP05 distribution and activity under differing oceanographic conditions.

We did, however, move this sentence further up within the paragraph (to lines 70-74).

Line 132: In “traveled already further westward”, I would remove the phrase “already further”.
Changed.

Figure 2: Some of the highest concentrations of SUP05 are in areas on the shelf that are 20-40 km from the coast and deeper than the elemental sulfur and denitrification maxima. These areas have no detectable sulfide, relatively little elemental sulfur, little or no evidence of denitrification, but some evidence of SUP05 carbon fixation. Does this provide evidence of a cryptic sulfur cycle? Meaning lots of SUP05, some carbon fixation, but no detectable sulfide.

Figure 2 shows the %SUP05 abundance, therefore the intensification shown in Figure 2d for SUP05 does not necessarily mean an equivalent increase in total abundance. In Figure 2f, the SUP05 CO₂ fixation rate scales with the SUP05 abundance, as the data are extrapolated from single cell measurements made at Stations L1 and U1. As the reviewer points out, these estimated rates at U2 & U3 are somewhat greater on the lower shelf. Nevertheless, elemental sulfur concentrations are still in the range of 100 to 200 nM at these depths, similar to the concentrations at L1, and likely support SUP05 activity.

Where these SUP05 populations evaluated by 16S rRNA gene analyses? Are they the same as the SUP05 in the metagenome?

See discussion below.

Lines 177-179: What about other SUP05, not Arctic96BD-19 or Peru upwelling SUP05? Are other species or strains potentially present that could have different metabolic capabilities. It would be helpful to know how good the probe is at discriminating all SUP05 from Arctic96BD-19. The data present suggest that Peru upwelling SUP05 are the dominant ecotype at U1 (30m), which is also the denitrification maxima and has the highest elemental sulfur concentrations. My comments above suggest that other SUP05 may be present. How do the authors know this group of SUP05 are the ones offshore? Also, the Peru upwelling SUP05 appear to have the same 16S rRNA gene sequences (Figure S1). So does the fact that the probe hits 10 SUP05 sequences really mean that it hits multiple copies of the same sequence?

The probe GSO131 was designed and tested on the latest available 16S rRNA dataset provided by the SILVA project (see M&M for details). Table S3 indicates the specificity and coverage of the GSO131 CARD-FISH probe. The GSO131 probe was designed to be specific for the dominant SUP05 ecotype at station U1 (also see Fig S1). The GSO131 probe used in combination with the two additional competitor oligonucleotides sequence exclude outgroup hits such as those related to the closely affiliated Arctic96BD-19 clade. To ensure the specificity of the probe we performed a double hybridization using the SUP05 GSO131 in addition to a probe designed to separately distinguish the Arctic96BD-19 clade (called GSO1290, which was designed in silico to target full-length 16S rRNA gene sequences recovered at Station 378, seen in Fig. S1). The double CARD-FISH result, shown in Figure S8 shows no detectable overlap between the two probes, as indicated by the two distinct hybridization signals. We are therefore confident that the specificity of the GSO131 probe is high and that no cross-hybridization is occurring

outside of the SUP05 clade based on the sequences highlighted in Fig S1. The multiple probe hits stem from very closely related SUP05, most likely from different strains of the same species

The same probe and competitor oligonucleotides applied at station U1 were used at offshore stations, thus reducing or eliminating Arctic96BD-19 and other cross-hybridization mismatches (See probe match for GSO131 Table S3).

Moreover, to test for the presence of an “indigenous” offshore population of SUP05 bacteria (i.e. a population distinct than the one identified in coastal waters) we performed a separate 16S rRNA gene analysis at an offshore station, station 378. Station 378, located further south of the eddy (coordinates - 13.75 °N -76.64 °E; Table S1), was non-sulfidic and had similar temperature-salinity characteristics as station L2, and was therefore not influenced by cross-shelf transport at the time of sampling. We attempted to recover SUP05 16S rRNA genes from waters within the OMZ at station 378 (83 m depth), however, were unable to amplify and recover SUP05 affiliated genes in our clone library preparations using the universal bacterial 16S rRNA gene primer set GM3f/GM4r.

To improve the detection limit and recoverability of full-length GSO sequences, we designed a more specific GSO forward primer GSO1f (designed in silico to target SUP05 and Arctic96BD-19 clades in the SILVA database) and used this in combination with the universal bacterial reverse primer GM4r. However, only sequences affiliated to Arctic96BD-19 bacteria were recovered, no sequence affiliated to SUP05 was found. The lack of recoverable SUP05 sequences (using two independent primer sets) reveals no evidence of an “indigenous” offshore SUP05 population.

Building on this we recovered 16S rRNA gene sequences from the metagenomic dataset at station L2 (now added to the phylogenetic tree (Fig. 3)), are indeed identical to SUP05 sequences recovered from Station U1. Notably the new GSO131 probe binds to all these sequences. Our results demonstrate that the GSO131 probe covers the SUP05 diversity in the Peru Upwelling, while excluding cross-hybridization with Arctic96BD-19 sequences. In addition, our phylogenetic 16S rRNA gene analysis in both inshore and offshore waters indicates that the Peru Upwelling, is dominated by a single SUP05 species originating from coastal sulfidic waters.

This text has been added as an extra discussion in the Supplemental Information file, in the Material and Methods. Furthermore, we have endeavored to make this point clear throughout the main text.

An additional note: There may be significant strain microdiversity that is not captured in our 16S rRNA gene analysis. ETSP SUP05 strain microdiversity analysis, although beyond the scope of this work, would be an exciting direction for future research.

Line 212: Are they correlated or not correlated? Are there any statistics? This is misleading. Especially since the next sentence (topic sentence of a paragraph) starts with “Despite such correlation”. This moves it from appears to is correlated. I don’t think this is accurate.

The reviewer is correct; the wording here is inaccurate. We have changed both sentences to clearly reflect the observation that measured rates of denitrification are broadly correlated with SUP05 cell densities at Line 230.

*“Denitrification rates also broadly correlate with SUP05 cell densities ($DN = \{1.14 * \log[\text{cell L}^{-1}]\} - 7.88$;*

$R^2=0.71$).

Furthermore, we have strengthened the opening sentence of the next section. This is important, because this is what is new in our paper.

“The presence and abundance of an organism in any given environmental setting, for instance SUP05 distributions in eddy influenced offshore waters, yields only limited information on the activity of the organism and its potential impact on the chemistry of the environment.”

Line 227: How does this compare with other estimates of carbon fixation in SUP05 (Hawley et al., 2014). *An interesting question and we appreciate that the reviewer raised this point. We went back and recalculated the estimates of carbon fixation made by Hawley et al. 2014, and, using the numbers in their final paragraph of their paper that derive from a model based on gene and protein abundances, came up with a range of 10 to 120 nmol C L⁻¹ day attributable to SUP05. Those estimates lie right in the same range as our measured SUP05 specific C fixation rates (2 to 409 nmol C L⁻¹ d⁻¹). Our experimental data provide solid support for estimates originally based on estimates of numbers of C fixation proteins derived from SUP05. This is an exciting convergence of experimental rate and –omics data.*

We have added a statement:

“Estimates of SUP05 carbon fixation made by Hawley et al. 2014 based on the abundances of SUP05 C fixation proteins range from 10 to 120 nmol C L⁻¹ d⁻¹ and lie in the same range as our experimentally determined in situ SUP05 specific C fixation rates (2 to 409 nmol C L⁻¹ d⁻¹).”

Line 258-259: Supports or does not support the cryptic sulfur cycle idea?

See initial comments on CSC. This sentence is a conclusion carefully drawn from our observations. It supports the idea that microbial sulfur metabolism is taking place and that it has a potentially important effect on N cycling. But that doesn't mean that it's part or not of a "cryptic cycle". We touch upon this theme then in the following paragraph (Lines 302-314) on sulfate reduction

Lines 260-262: Since SUP05 are well known nitrate reducing and sulfur oxidizing chemolithoautotrophs, I think the answer is clearly yes. It would be better to remove some of the straw man sentences and just state that SUP05 has important roles in marine carbon, nitrogen and sulfur cycling (ref).

This is not a "straw man" argument, rather a legitimate question. We have measured actively growing SUP05 cells, but the question still remains as to whether this is quantitatively relevant or not. Is this 1%? 10%? 100% of all denitrification? As it is, this sentence has disappeared in the process of rewriting this paragraph (Lines 283 and following).

Line 265: This is a big assumption because it assumes that coastal SUP05 and transported SUP05 are the same cells (Peru upwelling SUP05) with the same genetic potential and physiological responses to those measured at U1. How do published estimates for carbon fixation and nitrate utilization of SUP05 compare with the result reported in this manuscript (Hawley et al, 2016; Shah et al, 2016)?

Our CARD-FISH and metagenomic determinations demonstrate that these are the same organisms. Estimates from gene abundances or cell counts can only yield hypothetical estimates. Here we provide for the first time, direct measurements of SUP05 activity. Later we compare these to earlier estimates (see our previous comment on this point).

We compare single-cell rates of these organisms directly with the bulk rates of nitrate reduction (either to N₂ or to nitrite) and carbon fixation measured in the same waters. These results are presented in Table 1.

Lines 283-285: Against cryptic sulfur cycle?

See cryptic sulfur cycle comments above.

Line 290: All metagenomics data are from U1. Where 16S rRNA analyses conducted elsewhere and compared to those in the assembled genome? Meaning, how do the authors know these SUP05 are the same ones present elsewhere? If the new SUP05 probe specific to Peru upwelling SUP05 at U1 was used for single cell analyses at offshore sites, then it would be critical to know if these were the only SUP05 present at all other stations.

See prior discussion on this issue.

Line 303: Why oxygen, I don't see evidence of aerobic respiration in the figure or in the text.

The sentence has been changed accordingly and now reads (Lines 332-334):

“The presence of the high affinity cytochrome cbb3 complex indicates that electrons from the oxidation of reduced sulfur compounds can be used to reduce trace concentrations of dissolved O₂ to water.”

Line 305: The only complete genome sequence, *T. autotrophicus* is not included. Should probably reference either Shah and Morris 2015; Shah et al., 2016.

Thanks for catching this. Added.

Line 308: It's *T. autotrophicus* not *T. autotrophica*

Corrected

Line 311: See earlier comment regarding naming.

See above.

Line 314: This paragraph seems to be the argument against the cryptic sulfur cycle.

It is an argument against dissimilatory sulfate reduction as the only driver of cryptic sulfur cycling.

This is not an argument against the cryptic sulfur cycle. Rather it is an argument against local sulfate reduction being the only source of reduced sulfur. To make this clearer we now write:

“We cannot discount sulfide production from sulfate-reducing bacteria transported with SUP05, nonetheless, the large inventory of elemental sulfur at the shelf-water influenced “eddy” station L1 (>20

mmol m⁻² at L1; Fig. S3) can easily support SUP05 driven denitrification for weeks after transport of SUP05 bacteria and elemental sulfur into offshore ETSP waters“

Lines 350-353: This is where a lot of assumptions come together to make a statement that is somewhat misleading. This sentence suggests that a specific species of SUP05 capable of complete denitrification is the only group that is transported and that is active.

While our data demonstrates that the SUP05 clade bacteria ^UT. perditus is being transported and is active, we do not exclude other organisms being transported. Whether they remain active is another question. We have completely rewritten our concluding paragraphs to more clearly state that our data shows that cross shelf advection of sulfide oxidizing organisms and reduced sulfur (as elemental S) can support SUP05 activities in offshore waters. We have even changed the title of the section to “Implications of cross shelf transport for ‘cryptic sulfur cycling’. We thank the reviewer again for prodding us into being more definite.

Line 536-540: The growth factor was calculated at U1, 30m, where there is peak denitrification and peak denitrifying SUP05. This factor was then used to estimate values at other stations. This is where most of the assumptions come together. Another assumption is that sulfide, or elemental sulfur, is fully oxidized to sulfate. Do we know this? How do we know that the SUP05 populations are the same, and that the physiology of SUP05 cells are the same?

For the discussion on estimate of growth factor see discussion in Response Introduction. The identity of the SUP05 populations has been thoroughly commented on above.

It is indeed true that we do not know if sulfide or sulfur is fully oxidized to sulfate by SUP05. We have no direct single cell data available for sulfur cycling (no one does). But we do know that SUP05 here has the genetic capability to perform these reactions, and that they are assimilating CO₂. Moreover, unlike any previous study, these numbers are based on directly measured activities of a very well-characterized and delineated SUP05 population. In other words, these single-cell C assimilation rates directly link activity with identity, and we are thus able to produce a robust estimate of the importance of SUP05 to sulfur and nitrogen cycling in ETSP waters.

Reviewer #2 (Remarks to the Author):

This is a really nice paper and I suggest publication in its current form. Understanding the biogeochemical capacity of SUP05 is really important and this study does a fantastic job at (honestly) taking that task head-on.

There is a great accounting and involvement of documenting water mass movement and eddy development / propagation - this ends up being central to the story and is under-appreciated in previous work. Although the study, at its heart, is looking to understand the coupling between N and S, the calculations on environmental growth factors provide a nice check on the overall stoichiometry, and the

quantified contributions to the carbon cycle are a nice addition. In the end, the necessary reduced sulfur compounds are reportedly being transported via mesoscale eddies and not local “cryptic” sulfate reduction. This is super cool, and the authors should be clear that this actually goes against the logic reported in Canfield et al., where they call on local sulfate reduction and immediate sulfide oxidation. That is essentially the only criticism - the abstract is vague with respect to how these results impact earlier proposals.

We have rewritten the abstract to provide greater clarity as to how our results relate to earlier discussions in the literature. Please also see the discussion at the beginning of our Response on the topic of “cryptic sulfur cycling”. The point of “local” is a good one and we have tried to emphasize this in both the abstract and the conclusions. See also comments to Reviewer #1.

Finally, an additional sentence would be nice outlining the potential environmental role for thiosulfate, given that the genetic complement is present in SUP05. I presume that concentrations were below detection?

We indeed have thiosulfate data and have added a sentence into the text at Line 330:

“Thiosulfate was detected in the chemocline at Station U1 (400 nM), but was below detection (50 nM) at offshore stations L1 and L2 (Figure S5).”

The Methods and Materials section was likewise updated.

We have added a sentence pertaining to thiosulfate, and in the Methods and Materials.

Reviewer #3 (Remarks to the Author):

This manuscript describes an interesting set of field measurements of N, C, and S cycle processes in the ETSP. The manuscript is generally well-written, though some of the points were hard to tease out of the text. For example, I am unsure how to interpret the elemental sulfur concentrations given here: they are reported as if they’re measurements of dissolved S, but in fact must include substantial particulate S₀. Similarly, I didn’t learn until I got to the methods section that the rate measurements reported are really potential rates under anoxic conditions rather than estimates of what’s going on in the water column. This is an important distinction and really shouldn’t be buried in the methods section. On the whole, the findings are interesting, but the role of this cryptic S cycle in the water column isn’t clear to me given the experimental approach used.

The last point is well-taken. We have tried to clarify our stance on “cryptic sulfur cycling and throughout the point-by-point response both below in the responses to the other reviewers. Concerning potential rates, see our response below.

A number of more specific comments and suggestions follow.

Figure 1A: The black arrows don't show up very well against the dark blue. I suggest either adding a light border or using a different color altogether.

Changed.

Figure 1c: It would be helpful to have contours or some other indication of the depth of the water mass shown here. Even with that, I'm not sure that this panel adds anything meaningful to the paper.

We kept the sulfide plot in as it shows the rather narrow area where sulfide accumulates on the Peru shelf. We tried adding depth contours, but they muddied the figure and the contour looked like sulfide contour lines. We therefore decided to leave panel c without contours. However, we removed the mesoscale eddy in the panel c as it seems unnecessary to show it twice in two panels (i.e. panel b).

p. 6, first two paragraphs: The stations referenced should be clearly identified in Figure 1.

We have indicated the stations where we do in-depth analysis as white stars.

p. 6, paragraph 3: Elemental sulfur is highly insoluble in water and the μM S_0 concentrations reported are orders of magnitude higher than the equilibrium solubility. The analytical methods used to measure elemental sulfur concentrations (p. 16) will capture both dissolved and particulate S_0 , and the bulk of the reported concentrations must be accounted for by solid phase S_0 . Given the apparent importance of elemental sulfur in this story, I'm surprised at the lack of any effort to get at the solid/dissolved partitioning of S_0 , which might affect the movement of S_0 via mixing (e.g., p. 6, final paragraph) as well as the accessibility of S_0 to organisms.

The reviewer is correct that measured values of elemental sulfur here are much greater than the known solubility of cyclooctasulfur in seawater, which should be around 10 nM for these waters (Kamyshny, 2009, Geochim. Cosmochim. Acta). Colloidal elemental sulfur, however, is much more soluble.

Technically, what we have measured is not just dissolved cyclooctasulfur. We used the term "elemental sulfur" for convenience. It actually represents a number of possible zerovalent sulfur, including the sulfane component of inorganic polysulfides (especially in sulfidic waters), colloidal sulfur, sulfur attached extracellularly to organic matter and cells, and elemental sulfur deposited within cells. Even the elemental sulfur deposited within cells may exhibit a wide variability of forms, especially in an organic-rich milieu.

We have added the following text to the section (Line 146):

"Elemental sulfur" refers to zerovalent sulfur, mostly as cyclooctasulfur S_8 that including colloidal sulfur, S_8 externally associated with particles and cells, internal cellular deposits of S_8 , as well as the zerovalent component of dissolved inorganic polysulfides. "

The points that the reviewer raise concerning the speciation of elemental sulfur are excellent; but such an examination of the $\text{S}(0)$ speciation in these waters this is a topic for a future expedition.

p. 6, final paragraph: An estimate of the potential rate of supply of S_0 to the upper chemocline would

provide important context here. Specifically, can eddy diffusion provide enough S₀ to sustain meaningful rates of coupled denitrification?

Yes. We did this calculation and estimated an upward flux of S(0) of 6.6. mmol m⁻¹ d⁻¹. This would consume 7.9 mmol m⁻² d⁻¹ of NO₃⁻ based on Eq. 1. Much of the total nitrate consumption probably goes over S(0) oxidation in the chemocline. We have added this into the text at Lines 171-175..

“These estimates encompass the upward eddy diffusion flux of elemental sulfur in the chemocline, an upper limit for which we estimate to be -6.6 mmol S m⁻² d⁻¹.”

We have tried to limit the extent to which we put eddy diffusion calculations in the text, because they have far more uncertainty associated with them than the actual denitrification or single-cell 13C uptake rates. For instance, they assume steady-state and simple eddy diapycnal eddy exchange. The main point that we make is that there is more than enough nitrate to oxidize upward fluxes and S(0).

Figure 2: I suggest tweaking the color scales so that the highest concentrations are represented by the deepest red; it's visually confusing to have the scale transition to lighter colors for the highest concentrations.

Changed

Figure 2: Panel d shows that the bulk of the population of SUP05 is out on the outer shelf, while Panels e-f shows little or no activity associated with SUP05 out there. Why is the deeper population so inactive? Panel d shows the %DAPI as SUP05. The actual abundances of SUP05 (and activity) are lower.

See discussion of same point by Reviewer #1.

p. 7: This sort of calculation is exactly what I'd like to see for the postulated S₀ flux to the top of the chemocline.

See above.

p. 8, second paragraph: I don't see Stn L2 in Fig 2. Since this is an important reference station, it really needs to be shown and clearly identified

Station L1 and L2 occupy the same site and are only temporally separated. Figure 2 shows a composite of cross-shelf concentrations and activities that reflects the period where offshore transport dominated. We have now noted in the caption that L2 is at the same spot as L1:

“Figure 2: Distribution of concentrations, abundances and bulk and single cell activities in the Peru Upwelling OMZ as a function of distance from the coast. The composite plots show depth and cross-shelf distribution (a) nitrate, (b) dissolved sulfide, (c) elemental sulfur, (d) % total bacteria (DAPI) identified as SUP05, (e) bulk rates of denitrification, and (f) single-cell determined rates of CO₂ fixed by SUP05 at the time of eddy-induced offshore transport of shelf waters. Note that station L2 (not included in composite) was located near station L1, but occupied 11 days later. Black dots indicate sample depths

at each station included in the composite plots. “

p. 9, first paragraph: this is an important point, but it's not at all obvious from Figure 2. Stn L2 really needs to be added to that figure to make this point visually.

See aforementioned change to caption.

Table 1: I suggest explicitly showing the relative contribution of SUP05 to carbon fixation (about 65% if I'm reading the table correctly).

This is a good idea and we have included it into the newly revised Table 1.

Table 1: What is the value of depth-integrated SUP05 carbon fixation that's given in parentheses (8.6 ± 0.7)?

It was a placeholder for an earlier calculation that we forgot to remove. It has now been removed.

Table 1: For the final category, why not provide an estimate of the SUP05 contribution to the bulk rates. A quibble: these are areal rates, not turnover estimates.

The reviewer is correct. We have changed the wording in the new Table 1. .

Figure 3: For clarity, please be consistent in ordering of depths from left to right (L1 is deep-shallow while U1 is shallow-deep).

Changed.

p. 11, 2nd sentence: I understand that the rates at L1 and U1 are not significantly different, but isn't it interesting and worth noting that the specific rates are not just comparable, but actually tend to be higher at the offshore station?

Yes interesting, but as stated they are not significant and we would be very reluctant to read too much into slight differences.

p. 12, paragraph 2, sentence 3: What evidence is there that denitrification is primarily due to SUP05? Earlier, the authors estimate that up to 70% of the downward nitrate flux could be consumed by sulfide oxidation, presumably by SUP05 (p. 7, last paragraph). Since 70% is an upper limit, treating all of the denitrification as due to SUP05 will bias the estimates of this organism's impact on the N cycle.

The reviewer is correct that this introduces yet another large bias and more uncertainty in the estimation of environmental growth factor or yield. This is one of the reasons that we have backed away from this calculation and now employ laboratory/culture values for estimating the growth factor.

p. 13, first paragraph: What is the physical distribution of S in SUP05 cells? The nanoSIMS analysis should provide some insight into whether the S is stored in granules (e.g., S₀ granules), or in some other form.

It'd also be worth mentioning any association with either O or N, which could also provide clues to the physical form of the stored S.

The resolution of the nanoSIMS was not good enough to discern the form in which elemental S might be stored. Pure cyclooctasulfur will sublime under the high vacuum conditions that exist during nanoSIMS measurements. The fact that we still measure enriched values of sulfur in the SUP05 cells suggests that S is bound in a somewhat more stable matrix and not just as rhombic elemental S.

p. 17, second paragraph: Sparging with He will remove any O₂ present, which will promote anaerobic processes. This really needs to be brought up well before the methods section, since the measured rates of denitrification and anammox are going to be elevated relative to what's going on in the water column. The rates reported are potential, not in situ rates.

There have been lengthy discussions of this topic in other papers (Dalsgaard et al 2014, De Brabandere et al 2014 & 2012, Kalvelage et al 2013, Holtappels et al. 2012 and references therein). In fact, systematic studies using StoX sensors have shown that the issue is rather the opposite, that is the plastics associated with sampling (e.g. Niskin bottles) and the experiments introduce oxygen (De Brabandere et al 2012). We have put in a sentence to this effect in the Methods.

=====End of Review=====

Reviewer #4 (Remarks to the Author):

This is an exciting and well designed study that extends our understanding of the ecology and physiology of the globally important sulfur-oxidizing bacterial SUP05 group. Set in the highly dynamic Peru upwelling region, this work demonstrates changes in the physiology and activity of the SUP05 through the influence of physical and chemical oceanographic processes while also shaping the cycling of C, N and S within the oxygen minimum zone. The manuscript is coherently written, quantitative, and incorporates compatible, state-of-the-art methods. I have only a few questions, minor comments and suggestions for further strengthening this manuscript.

Line 230: If the SUP05 represented ~65% of the total dark CO₂ fixation, did you see evidence of other cells in addition to those targeted by the SUP05 probe that showed active ¹³C-bicarb incorporation by nanoSIMS?

Indeed, we saw other bacteria that could contribute to CO₂ fixation, in particular numerous Epsilonproteobacteria on the shelf, including Arcobacter, were also present at some shelf stations. Their physiology and impact is the subject of another study.

Similarly,

Lines 255-256: Can you clarify this comment about station L2? does this mean there are other organisms contributing to dark C fixation in addition to SUP05, or is there very low rates of C fixation overall?

SUP05 cell densities and its single-cell rates of CO₂ fixation were lower as compared to station L1; and SUP05 made only a minor (8%) contribution to CO₂ fixation. While other bacteria might be involved in CO₂ fixation, volumetric rates below the euphotic zone were at, or approaching, the limit of detection at station L2. Thus, bulk CO₂ fixation rates at station L2 even by other bacteria could be considered insignificant compared to other stations which had two and six-fold higher depth integrated CO₂ fixation rates (stations L1 and U1, see Table 1).

We added the following text (Line 279):

“Our estimates suggest that that SUP05 bacteria are responsible for 65 to 134% of the dark carbon fixation rates on the shelf and at the eddy-influenced L1 station, but only 8% of the dark carbon fixation at Station L2 (Table 1). Furthermore, dark carbon fixation at the offshore, non-eddy station is low.”

Line 300: From the nanoSIMS data was there any visible difference in the S distribution associated with the SUP05 cells? Any extracellular precipitation observed, or just intracellular S/C ratios?

Again, were any other bacteria analyzed from these samples and do you have data on their S/C ratio as a point of comparison (i.e. is this elevated S/C ratio unique to SUP05 or is there higher S/ C for other cells in the same sample)?

Some epsilonproteobacteria at Station U1 have somewhat greater S/C ratios, although interestingly Arcobacter (another sulfide oxidizer found at Station U1) exhibits consistently lower S/C ratios. This may indicate something about the type of elemental sulfur and means of storage. Such a discussion goes beyond the scope of this paper, and we are working on presenting it in another manuscript. Other non-sulfur oxidizing bacterial cells also have lower values.

Line 305-306: is this statement based on the genome prediction or your activity based measurements? If associated with the genome data, I suggest rewording the sentence indicating this is a predicted genomic capability rather than stating that they are capable of complete denitrification coupled with sulfide and elemental sulfur oxidation. From the data presented in this study, none of the specific physiology experiments directly indicates their ability to use elemental sulfur.

We agree, and have rephrased the sentence accordingly (Line 340).

“Thus, the genomic capability of SUP05 organisms active in the ETSP predicts that they can perform complete denitrification coupled to sulfide and elemental sulfur oxidation.”

Line 310: do you have an ANI percentage you can provide for the similarity between T. autotrophica and

peru upwelling SUP05 genome bin?

Our bin has an ANI of 74% to Thioglobus autotrophicus, which clearly makes it a different species.

Line 438: Need to provide information about which samples were prepared for DNA extraction/metagenomics. How was the sample concentrated and what was the volume?

DNA was extracted from 1-2 liters of water sample concentrated on a polycarbonate filter with 0.2 μm pore-size, and is included in the Materials and Methods (Lines 470 and following).

Lines 478-480: Was the specificity of the newly designed GSO131 probe tested aside from in silico predictions? If so, add a comment here. Also given that this is a new probe, if you have any information about optimization that would also be good to include (beyond the footnote in Table S2).

The GSO131 probe was tested both in silico and then evaluated on filters collected at station U1 using different formamide contents (10%, 20%, 30%, 40%, 50%, and 60%). We have clarified this (Line 526) and in the Supplemental Information.

Line 500: The CARD-FISH procedure includes a methanol /H₂O₂ treatment and possibly a ethanol dehydration series which can leach elemental sulfur inclusions from cells- do you have any sense of how this may have impacted your S/C ratio data and does this have implications for the form of sulfur observed in the cells if it wasn't removed during the dehydration step?

This is a good point. We may also lose some sulfur during the ethanol drying step. As we indicated earlier by Reviewer #3, sublimation under high vacuum may affect cyclooctasulfur deposits. Thus, these are conservative estimates of the sulfur content. See the response to Reviewer #3 for more details.

Line 502: Did you use the Oregon green 19F signal to confirm the hybridization signal by nanoSIMS? If so, please include that information here and cite a reference.

The reviewer is correct we did use the 19F signal to detect SUP05 cells hybridized with the GSO131 probe. We therefore have included this information along with a reference at Line 550).

Line 507: What was the pA value for the Cs⁺ ion beam during the analysis?

1.5 – 2 pA ; included at Line 555.

Some information about cellular ¹⁵N data would be worth commenting on in the text given that this was one of the isotope ratios measured- even in the absence of a trend, it would be helpful to know what that data looked like. Also some discussion on variation in secondary ion production for sulfur vs carbon is warranted in the context of interpretation of S/C cell ratios.

We thank the reviewer for the comment we have therefore added more information regarding the single-cell SUP05 assimilation of ¹⁵N-NO₃⁻ in the manuscript. We find that SUP05 cells at station U1 (60 m depth) were indeed enriched in ¹⁵N-NO₃⁻ over the background reaching nitrogen assimilation rates of

(0.002-0.015 fmol cell⁻¹ d⁻¹), which increased linearly with rates of carbon fixation (Fig. S8; R = 0.70, p<0.05 Pearson correlation analysis). We have added this plot to the SI (Figure S6), along with a brief description of the single-cell nitrogen assimilation rate calculation in the M&M and a short sentence in the revised manuscript at Line 246.

In secondary ion mass spectrometry sulfur shows a better ionization potential than carbon. Ionization of various elements is also matrix dependent. Nevertheless, the matrices across all the cell samples were similar, and certainly across the cells identified as belonging to the SUP05 clade. We have included this comment also in the Materials and Methods.

Figure 2. Much of the data is associated with station L2, but this station is not included in the figure. *Station L1 and L2 occupy the same site and are only temporally separated. Figure 2 shows a composite of cross-shelf concentrations and activities that reflects the period where offshore transport dominated. We have now noted in the caption that L2 is at the same spot as L1. See new Figure 2 caption.*

Reviewers' comments:

Reviewer #1 (Remarks to the Author):

The authors of "Oxygen minimum zone cryptic sulfur cycling sustained by offshore transport of key sulfur oxidizing bacteria" have done an excellent job revising the manuscript and have addressed the majority of my concerns. More specifically, they have done a very nice job describing their results in the context of the cryptic sulfur cycle by providing an alternative explanation for a novel observation. Also, my concern regarding the phylogeny was directed more at the use of "Candidatus", which in my mind means that there is a cultured representative. The solution of naming the Peru species "Uncultured" Thioglobus perditus is excellent and it is now clear.

My only remaining concern is that the carbon fixation rates are based on cell size and are likely too high. The authors indicate that their estimates are within the range of previously reported data (7-260 fg C/cell), citing Fukuda et al., 1998. This is not totally accurate. The higher value comes from Table 4, which references Kroer 1993. This paper reported higher numbers after incubating natural populations and observing significant increases in cell size. The direct measurements reported by Fukuda et al., 1998 (Table 3) average 15.7-47.9 fg C/cell in coastal systems.

Overall, this is an excellent paper and I don't think this should preclude publication. However, I encourage the authors to consider publishing a range. As is, their carbon content/cell could overestimate carbon fixation by a factor of 1.6 to 4.8, depending on the measured carbon content/cell reported by Fukuda et al., 1998 for coastal systems.

Some smaller details:

1. I think GSO stands for gamma sulfur oxidizers, rather than gamma sulfide oxidizers, but I could be wrong. Double check.
2. Line 221: "sulfid" should be "sulfide"

Reviewer #3 (Remarks to the Author):

This manuscript describes an interesting set of field measurements of N, C, and S cycle processes in the ETSP. As in the previous submission, the manuscript is generally well-written, but I remain concerned about some of the linkages among elemental cycles postulated in this manuscript. On a general level, I worry that the authors are pushing the data as far as they can, which I don't think is always the most productive way to advance understanding. I'm perhaps (likely) a minority view in this, but I think that the current manuscript requires substantial reworking, particularly in its numeric parts, before it should be considered for publication.

A number of more specific comments and suggestions follow (all lines are in the marked up pdf showing changes from the earlier version of the ms.).

Lines 52-69: this is a much more compelling framing of the problem than in the original manuscript – the last sentence is critical in providing context and setting the stage for the rest of the paper.

Lines 66-67: it might be worth noting that this is exactly equivalent to the coupled nitrification-denitrification process that commonly occurs in marine sediments.

Lines 90-94: This is a good null hypothesis and could be presented as such.

Lines 101-103: another critical piece of context that is clearly presented in this revision

Lines 107-111: These two sentences raise an obvious question – if there's no geochemical evidence for sulfate reduction, then why does it matter that the metabolic capacity is present? There are lots of examples of organisms capable of carrying out metabolic pathways that are incompatible with the environment they're found in, so it would be worth developing this observation a bit more here in the introduction.

Figure 1a: the black arrows are very hard to see against the dark blue background. I suggest using white arrows or a broader white line around the dark arrows.

Figure 1b & 1c: In my version, the stars and circles look very similar on screen, particularly nearshore (e.g., the station furthest to the NE). I recommend using either colors or different symbols to distinguish them.

Lines 178-181: This is a welcome clarification of what was actually measured.

Lines 195-202 (?): The inserted text does a good job of clarifying where Stn L2 is located.

Line 20?: in the 2nd line of the paragraph below the figure legend, "of" is missing after "...eddy diffusion coefficient...". The line numbers are hashed in this section of the pdf I'm looking at.

Lines 209-210: Although there's sufficient nitrate to oxidize all the sulfide that's moving upwards, how can you rule out oxygen as the electron acceptor? Oxidation of sulfide with molecular oxygen is highly exergonic, so I don't see why this reaction is ignored here.

Lines 247-252: This certainly implies that S₀ is being passively transported offshore and not being generated in offshore waters.

Lines 267-268: Providing the correlation coefficient is important here – it really reinforces the point that SUP05 appears to be involved.

Lines 280-282: What is the true specific rate of C fixation (units of inverse time)? Although the volume-specific rates are interesting and important, the specific rates are a much better indication of the overall importance of a pathway.

Lines 282-288: The rates reported here and those of Hawley et al. appear to differ by at least a factor of 4! The text as written appears technically correct given the broad range of rates reported in the current manuscript, but it's deeply misleading to claim that the rates of 10-120 nmol/L/d reported by Hawley et al. are the same as a mean of 409 ± 50 nmol C/L/d! This is the sort of apparently quantitative (but false) statement that makes me deeply suspicious of a manuscript. It's probably an innocent oversight/overstatement, but it really makes me worry about every other claim made in the manuscript, which is a real shame....

Line 291: Given my reservations about the way rates are being reported/handled, I suggest rewording this to "...shows that SUP05 COULD contribute UP TO 65% of the bulk dark C fixation..."

Lines 317-322: This is beginning to sound circular – we're to believe that SUP05 is active and important because they're present and perturbation experiments show that they are active, but the claim of activity is based on perturbation experiments

Line 344: True, but this should be placed in the context of other potential denitrifiers that are present in the system. I'm troubled by this sort of argument, which really relies on absence of information to make a case for a particular process; I would be much more sympathetic if some estimate of how the putative SUP05 contribution scaled relative to that of other denitrifiers were presented.

Lines 351-352: what evidence is there that sulfate reduction is occurring locally? Earlier, we saw evidence that SO might be present offshore because of advection, so this seems like a stretch.

Lines 359-363: OK, this gets to my point in the previous note, but this is important context that should be presented before what I view as the speculative text on the role of local sulfate reduction.

Line 396: Couldn't this also reflect an inability to metabolize sulfur, leading to intracellular accumulation?

Lines 414-422: I understand the desire to name the microbe, but this seems a stretch given the absence of data directly showing its role in denitrification!

Lines 446-447: at any one time, I presume...

Lines 461-462: I'm confused by the invocation of cryptic S cycling here. Why is this needed/important? My sense from the rest of the manuscript is that advection is key in moving reduced S and SUP05 offshore, but that there's no unequivocal evidence that SUP05 activity offshore is quantitatively important.

=====
=====End of Review=====

Reviewer #4 (Remarks to the Author):

I appreciate the effort the authors made in revising the manuscript and including new data. The manuscript is much improved and I am satisfied with their responses to the concerns raised by the reviewers.

Reviewers' comments:

Reviewer #1 (Remarks to the Author):

The authors of "Oxygen minimum zone cryptic sulfur cycling sustained by offshore transport of key sulfur oxidizing bacteria" have done an excellent job revising the manuscript and have

addressed the majority of my concerns. More specifically, they have done a very nice job describing their results in the context of the cryptic sulfur cycle by providing an alternative explanation for a novel observation. Also, my concern regarding the phylogeny was directed more at the use of “Candidatus”, which in my mind means that there is a cultured representative. The solution of naming the Peru species “Uncultured” *Thioglobus perditus* is excellent and it is now clear.

My only remaining concern is that the carbon fixation rates are based on cell size and are likely too high. The authors indicate that their estimates are within the range of previously reported data (7-260 fg C/cell), citing Fukuda et al., 1998. This is not totally accurate. The higher value comes from Table 4, which references Kroer 1993. This paper reported higher numbers after incubating natural populations and observing significant increases in cell size. The direct measurements reported by Fukuda et al., 1998 (Table 3) average 15.7-47.9 fg C/cell in coastal systems.

Overall, this is an excellent paper and I don't think this should preclude publication. However, I encourage the authors to consider publishing a range. As is, their carbon content/cell could overestimate carbon fixation by a factor of 1.6 to 4.8, depending on the measured carbon content/cell reported by Fukuda et al., 1998 for coastal systems.

The reviewer has touched upon an important point. We agree that the carbon contents based on cell size are high relative to those reported in Fukuda et al. (1998) and to most cells in coastal systems. The reviewer's comments encouraged us to revisit the original nanoSIMS data used for calculating cell volumes, and we see that where we have erred on the high side is that we simply calculated the cell volumes based on a sphere. In reality the cell shapes on the nanoSIMS filters varied in form from coccoid to short thick rods to prolate spheroids. Cylinders and prolate spheroids will yield lower volumes (by 42 to 49% for a 1.4 length-width aspect) for a cell with a length of 0.81 μm . We have now conservatively recalculated the cell volume treating each cell as a cylinder with rounded half spheres (see Lines 543-549 in Materials and Methods). From these estimates we obtain cell volumes of 0.18 to 0.39 μm^3 (Table 1). The fit by Romanova and Sazhin (2010) that we used also takes into account that larger cells have a lower carbon density than smaller cells. Thus, the volumes of SUP05 measured yield a range of carbon contents from 61 to 83 fgC, which are 21% lower on average than our previous estimate. Despite this downward correction, the SUP05 cells observed in the Peru Upwelling still have cell volumes and C contents well above those typical for marine bacteria.

We thank the reviewer for encouraging us to more closely consider the problem of the relationship between cell size and carbon (and other elements). Considerations and discussions of uncertainty are now explicitly described in the section on SUP05 contribution to bulk dark carbon fixation (Lines 237-251). As we state in the text, the carbon content and carbon density of living bacterial and archaeal cells that go into such calculations are points of considerable discussion and uncertainty. We believe that we have clearly addressed this problem, and hope that our work encourages greater attention to this issue.

Some smaller details:

1. I think GSO stands for gamma sulfur oxidizers, rather than gamma sulfide oxidizers, but I could be wrong. Double check.

The reviewer is correct and this figure has been changed.

2. Line 221: “sulfid” should be “sulfide”

Changed.

Reviewer #3 (Remarks to the Author):

This manuscript describes an interesting set of field measurements of N, C, and S cycle processes in the ETSP. As in the previous submission, the manuscript is generally well-written, but I remain concerned about some of the linkages among elemental cycles postulated in this manuscript. On a general level, I worry that the authors are pushing the data as far as they can, which I don't think is always the most productive way to advance understanding. I'm perhaps (likely) a minority view in this, but I think that the current manuscript requires substantial reworking, particularly in its numeric parts, before it should be considered for publication.

We can certainly understand the concerns of the reviewer on this point. The right balance between reporting observations (in the form of depth distributions of compounds, cells and rates) and the request to put the data into perspective (i.e., what does this mean quantitatively for the cryptic sulfur cycle?) needs to be made. We thank the reviewer for confronting us with this point.

Therefore we have rewritten the section on *Single-cell activities of SUP05 Bacteria* (Lines 214-290). In doing so, we proceed from hard data from our single-cell experiments, to SUP05 dark carbon fixation rates, to broad estimations of how might SUP05 specific rates compare to bulk carbon fixation, denitrification and sulfide oxidation rates. Accompanying each calculation we are careful to state assumptions and caveats. We hope that this reworking of the text, as well as the description of the calculations, will allow readers to clearly appreciate the power, as well as limitations, of our single-cell approach.

A number of more specific comments and suggestions follow (all lines are in the marked up pdf showing changes from the earlier version of the ms.).

Lines 52-69: this is a much more compelling framing of the problem than in the original manuscript – the last sentence is critical in providing context and setting the stage for the rest of the paper.

Lines 66-67: it might be worth noting that this is exactly equivalent to the coupled nitrification-denitrification process that commonly occurs in marine sediments.

This may indeed be true with respect to the formation of N₂O, and it is interesting in and of itself, but is not directly relevant to the discussion on the physiology of SUP05. Discussing nitrification-denitrification in sediments would unnecessarily distract from the main point here.

Lines 90-94: This is a good null hypothesis and could be presented as such.

We considered presenting this point as a null hypothesis, but we felt that this distracted from the other issues concerning SUP05 (e.g. accuracy of the FISH probe, metabolism, relationship between presence and activity). Nevertheless, we have moved it down in the paragraph, where we believe it reads better (at Lines 84-89). It also fits better with our response to the reviewers comment at Lines 107-111.

Lines 101-103: another critical piece of context that is clearly presented in this revision

Lines 107-111: These two sentences raise an obvious question – if there's no geochemical evidence for sulfate reduction, then why does it matter that the metabolic capacity is present? There are lots of examples of organisms capable of carrying out metabolic pathways that are incompatible with the environment they're found in, so it would be worth developing this observation a bit more here in the introduction.

We agree with the latter statement, and raise exactly this point later in the discussion. On the other hand, just because there is no geochemical evidence for sulfate reduction, this does not mean that the process does not take place. This is the definition of "cryptic cycling". The sulfate oxygen isotope data presented by Johnston et al. (2014) place upper limits on the rates of cryptic sulfur cycling, but still may be too insensitive to detect the rates observed by Canfield et al., (2010). This is very much part of the controversy about the 'cryptic sulfur cycle'. We have amended the sentence to make this clear (Lines 81-84).

"Geochemical evidence that would point to substantial rates of microbial sulfate reduction in offshore waters has not been found in the ETSP, but these natural abundance stable isotope measurements may still be too insensitive to detect estimated rates of sulfur cycling in these OMZ waters."

Figure 1a: the black arrows are very hard to see against the dark blue background. I suggest using white arrows or a broader white line around the dark arrows.

Improved.

Figure 1b & 1c: In my version, the stars and circles look very similar on screen, particularly nearshore (e.g., the station furthest to the NE). I recommend using either colors or different symbols to distinguish them.

Improved.

Lines 178-181: This is a welcome clarification of what was actually measured.

Lines 195-202 (?): The inserted text does a good job of clarifying where Stn L2 is located.

Line 20?: in the 2nd line of the paragraph below the figure legend, "of" is missing after "...eddy diffusion coefficient...". The line numbers are hashed in this section of the pdf I'm looking at.

Changed.

Lines 209-210: Although there's sufficient nitrate to oxidize all the sulfide that's moving upwards, how can you rule out oxygen as the electron acceptor? Oxidation of sulfide with

molecular oxygen is highly exergonic, so I don't see why this reaction is ignored here. The reviewer is correct in that molecular oxygen is still an excellent, if not necessarily the preferred electron acceptor under microaerophilic conditions. The main point of the discussion here, however, is to point out that the nitrate flux into the chemocline is more than enough to account for all of the sulfide consumption via oxidation taking place there. Microorganisms that facilitate this reaction, e.g. *U. perditus*, should dominate the chemocline community. There may also be a flux of dissolved oxygen into the chemocline, but as seen in Figure S4, the oxygen gradient is miniscule in comparison to nitrate. We have been careful to frame our discussion here, with qualifiers such as "upper limits" and "up to 70% of the total nitrate flux could be attributed to...". (Lines 157-159)

As we stated in our first response to the reviewers "We have tried to limit the extent to which we put eddy diffusion calculations in the text, because they have far more uncertainty associated with them than the actual denitrification or single-cell 13C uptake rates. For instance, they assume steady-state and simple eddy diapycnal eddy exchange. The main point that we make is that there is more than enough nitrate to oxidize upward fluxes and S(0)."

Lines 247-252: This certainly implies that S₀ is being passively transported offshore and not being generated in offshore waters.

Yes, and this is also why we included the Temperature-Salinity-Sulfur plot in the SI.

Lines 267-268: Providing the correlation coefficient is important here – it really reinforces the point that SUP05 appears to be involved.

Lines 280-282: What is the true specific rate of C fixation (units of inverse time)? Although the volume-specific rates are interesting and important, the specific rates are a much better indication of the overall importance of a pathway.

This is a good point and we have added this into Table 1. We have also made the calculation in the Methods and Materials Section more explicit (Lines 559-566).

Lines 282-288: The rates reported here and those of Hawley et al. appear to differ by at least a factor of 4! The text as written appears technically correct given the broad range of rates reported in the current manuscript, but it's deeply misleading to claim that the rates of 10-120 nmol/L/d reported by Hawley et al. are the same as a mean of 409 ± 50 nmol C/L/d! This is the sort of apparently quantitative (but false) statement that makes me deeply suspicious of a manuscript. It's probably an innocent oversight/overstatement, but it really makes me worry about every other claim made in the manuscript, which is a real shame....

We believe that our statement concerning the Hawley et al. 2014 paper has been misunderstood. We originally wrote:

"Estimates of SUP05 carbon fixation made by Hawley et al. 2014 (12) based on the abundances of SUP05 C fixation proteins range from 10 to 120 nmol C L⁻¹ d⁻¹ and lie in the same range as our experimentally determined in situ SUP05 specific C fixation rates (2 to 409 nmol C L⁻¹ d⁻¹). "

The Hawley et al., 2014 estimates do fall within a comparable range of our experimentally determined estimates that cover the entire range of stations that we investigated where *T. perditus* is abundant and active. If there was an oversimplification made, it was that we did not use the full range of values of the ranges given in Table 1 (now recalculated to be 1.3 to 592 nmol C L⁻¹ d⁻¹; see Response to Reviewer 1). We have changed this to indicate the full range.

Apparently, we have not made it sufficiently clear that we were comparing our SUP05 specific data on C fixation with the completely different approach taken by Hawley et al.. The numbers for C fixation by SUP05 from Hawley et al., 2014 are based on protein content and qPCR data from Saanich Inlet coupled with a numerical model, which they then extrapolate using the global volume of OMZs. In the first round of reviews, Reviewer 1 asked us how our directly determined single-cell estimates compared with those of Hawley et al.. We agreed that this was an interesting idea to include, and indeed, comparisons show that our local, direct determined rates and those taken from a Hawley et al.'s *-omics* approach converge. Given the differing sites, approaches and scales, we find this convergence to be very encouraging (for both the proteomics and single-cell experimental communities).

We have rewritten the discussion of the Hawley et al., (2014) paper. It now reads (Lines 259-265):

"It is interesting to note that, Hawley et al. (12), in extrapolating from a gene-centric biogeochemical model based on qPCR data and the abundances of SUP05 C fixation proteins in Saanich Inlet, estimated SUP05 associated C fixation rates of 10 to 120 nmol C L⁻¹ d⁻¹ for OMZ waters. Our calculated in situ rates of SUP05-specific carbon fixation rates for the ETSP OMZ waters (ranging from 1.3 to 592 nmol C L⁻¹ d⁻¹ for stations U1 and L1; Table 1) confirm the proposed importance of SUP05 to dark carbon fixation in shelf and sulfur-rich OMZ waters."

Line 291: Given my reservations about the way rates are being reported/handled, I suggest rewording this to "...shows that SUP05 COULD contribute UP TO 65% of the bulk dark C fixation..."

This has been changed and as noted earlier, we have made extensive effort to show how these values are scaled-up, and the caveats associated with these calculations.

Lines 317-322: This is beginning to sound circular – we're to believe that SUP05 is active and important because they're present and perturbation experiments show that they are active, but the claim of activity is based on perturbation experiments

We agree that the placement of this sentence may cause some confusion. We believe that this confusion has been eliminated in the rewriting of this section.

Line 344: True, but this should be placed in the context of other potential denitrifiers that are present in the system. I'm troubled by this sort of argument, which really relies on absence of information to make a case for a particular process; I would be much more sympathetic if some

estimate of how the putative SUP05 contribution scaled relative to that of other denitrifiers were presented.

There are indeed other nitrate reducing and sulfide oxidizing organisms in these waters, and we make this point now in the text (Lines 251-253). The manuscript is concerned the micro-organism that has been identified as a key organism in the cryptic sulfur cycle, i.e. SUP05. Moreover, it is not presently possible to quantitatively assess at the single-cell the contribution of organisms such as heterotrophic denitrifiers, which we state in Lines 266-269:

“Growing, or actively autotrophic SUP05 cells would also be expected to have an impact on denitrification and sulfide oxidation. In contrast to the determination of SUP05-specific C or N assimilation using nanoSIMS, the direct experimental determination of single-cell or single clade respiration rates in the environment is not yet possible.”

As we state now in the text (Lines 267-269), the only path to getting single-cell or clade-level respiration data is via single cell C assimilation data. For most denitrifying bacteria and many sulfide oxidizers, such single-cell data is not possible. Fortunately, SUP05 is a very active autotroph at Stations U1 and L1.

Lines 351-352: what evidence is there that sulfate reduction is occurring locally? Earlier, we saw evidence that S0 might be present offshore because of advection, so this seems like a stretch.

Lines 359-363: OK, this gets to my point in the previous note, but this is important context that should be presented before what I view as the speculative text on the role of local sulfate reduction.

We revised the opening sentence of this paragraph to make to take care of any misunderstandings. Please note that we also moved this paragraph into the section on SUP05 distributions, where we believe it fits better.

Line 396: Couldn't this also reflect an inability to metabolize sulfur, leading to intracellular accumulation?

Yes, technically might be true. But the simplest hypothesis is that it represents accumulation of sulfur that can continue to be oxidized further to sulfate (based on its genome).

Lines 414-422: I understand the desire to name the microbe, but this seems a stretch given the absence of data directly showing its role in denitrification!

We concur and have changed this to (Lines 324-325):

“^UThioglobus perditus”(45). Perditus means lost. The Peru Upwelling SUP05 bacterium ^UT. perditus finds itself lost in the offshore OMZ waters.

Lines 446-447: at any one time, I presume...

Changed.

Lines 461-462: I'm confused by the invocation of cryptic S cycling here. Why is this needed/important? My sense from the rest of the manuscript is that advection is key in moving reduced S and SUP05 offshore, but that there's no unequivocal evidence that SUP05 activity offshore is quantitatively important.

We do believe that this is important to state, as did the other reviewers. We have unequivocal evidence that *T. perditus* is active in offshore waters (i.e. at Station L1). Further data on the genome and sulfur contents indicates that this activity may be linked to the oxidation of co-transported elemental sulfur to sulfate. While it is not the same the 'cryptic sulfur cycle' envisaged by Canfield et al., it does constitute sulfur cycling offshore. To make this clear we have amended the sentence at Line 342-345.

"Our results show that mesoscale eddy driven water mass movement can explain the abundance and activity of sulfide oxidizing bacteria in sulfide-poor offshore OMZ waters and can drive 'cryptic sulfur cycling' via the continued oxidation of co-transported elemental sulfur."

=====End of Review=====

Reviewer #4 (Remarks to the Author):

I appreciate the effort the authors made in revising the manuscript and including new data. The manuscript is much improved and I am satisfied with their responses to the concerns raised by the reviewers.

Thank-you.

REVIEWERS' COMMENTS:

Reviewer #1 (Remarks to the Author):

The authors have addressed my concerns regarding cells size. I'm not convinced that their estimates of carbon content/cell are correct, but they have done some additional work to come up with more accurate estimates of cell size.

The authors do need to correct the NCBI file for "U Ca. Thioglobus perditus" to clearly indicate that it is uncultured. The genome is now available, but it is impossible to tell from the metadata that this genome was assembled from metagenomic data and that this genome represents an uncultured organisms. The file, in fact, even designates the strain, which implies that this is a culture. This is a dangerous precedent to set and needs to be corrected prior to publication. Specifically, the source of DNA should be metagenomic, the name should state "uncultured" and the strain ID should be removed.

Otherwise, I support publication of the manuscript.

Reviewer #3 (Remarks to the Author):

This manuscript describes an interesting set of field measurements of N, C, and S cycle processes in the ETSP. I actually reviewed this version twice to try to avoid biasing myself – first without reading the response to reviewers and my previous reviews, and then again with that additional context in hand. On both reviews, I enjoyed the manuscript, which like its predecessors is generally well-written. In addition, the revisions made in preparing the current version have significantly improved and clarified it, and address all of the concerns I had with the earlier iterations. This will be a widely read and cited manuscript of broad interest to oceanographers and biogeochemists.

A number of more specific comments and suggestions follow.

Lines 38-51: this introductory paragraph does a nice job of framing the question in a compelling way – the sentence added to the end helps provide context and motivation for the rest of the paper.

Lines 77-80: an important point that is often overlooked!

Line 114: a minor point, but "curl" may provide the wrong connotation to physically/mathematically inclined readers. "veering" may be a better choice here.

Lines 135-138: a very helpful explanation of the presence of elemental sulfur at concentrations well above its solubility.

Lines 146-147: This sentence implies that N₂ production will occur only at the top of the chemocline. Couldn't this occur throughout the chemocline wherever both NO₃⁻ and S₀ are present?

Lines 161-213: This section is very clearly written and makes a strong case for the potential importance of SUP05 and the cryptic sulfur cycle. A variety of important contextual information appears to have been added in this revision, and I appreciate the work done to temper some of the statements I thought were too strong in the previous versions.

Lines 214 ff: this rewritten section is very clear and compelling.

Response to Reviews

March 21, 2018

Callbeck et al., *Oxygen minimum zone 'cryptic sulfur cycling' sustained by offshore transport of key sulfur oxidizing bacteria*

We wish to thank the reviewers for their thorough, thought-provoking and constructive reviews that have led to an improved manuscript. We sincerely appreciate the time and effort that they have put into the various discussions on the data, concepts and presentation.

Our replies and corrections with respect to the last set of reviews are indicated in italics below.

REVIEWERS' COMMENTS:

Reviewer #1 (Remarks to the Author):

The authors have addressed my concerns regarding cells size. I'm not convinced that their estimates of carbon content/cell are correct, but they have done some additional work to come up with more accurate estimates of cell size.

As discussed in the previous reviews, we believe that we have addressed this issue as well as possible, and we believe that our estimates of the cell volume – carbon content relationships are the most accurate. Obviously, this is an issue that needs to be addressed within the marine biogeochemical community, and we thank the reviewer again for making us think more carefully about this issue.

The authors do need to correct the NCBI file for "U Ca. Thioglobus perditus" to clearly indicate that it is uncultured. The genome is now available, but it is impossible to tell from the metadata that this genome was assembled from metagenomic data and that this genome represents an uncultured organisms. The file, in fact, even designates the strain, which implies that this is a culture. This is a dangerous precedent to set and needs to be corrected prior to publication. Specifically, the source of DNA should be metagenomic, the name should state "uncultured" and the strain ID should be removed.

We thank the reviewer for taking the effort to check on the NCBI file and point this out. We have contacted NCBI and have taken steps to correct the file. "Strain" has been changed to "isolate" and the source is indicated as being "marine metagenome". We are not entirely happy with the term "isolate" and we have also tried to convince them to use "Uncultured" in the name. Their reply was that "Candidatus" should suffice.

Given the discussion during the review process, however, and our effort to use the "U" indicator for uncultured organism after Konstantinidis et al., 2017 we are still endeavoring to get the NCBI to alter the name.

Otherwise, I support publication of the manuscript.

Reviewer #3 (Remarks to the Author):

This manuscript describes an interesting set of field measurements of N, C, and S cycle processes in the ETSP. I actually reviewed this version twice to try to avoid biasing myself –

first without reading the response to reviewers and my previous reviews, and then again with that additional context in hand. On both reviews, I enjoyed the manuscript, which like its predecessors is generally well-written. In addition, the revisions made in preparing the current version have significantly improved and clarified it, and address all of the concerns I had with the earlier iterations. This will be a widely read and cited manuscript of broad interest to oceanographers and biogeochemists.

A number of more specific comments and suggestions follow.

Lines 38-51: this introductory paragraph does a nice job of framing the question in a compelling way – the sentence added to the end helps provide context and motivation for the rest of the paper.

Lines 77-80: an important point that is often overlooked!

Line 114: a minor point, but “curl” may provide the wrong connotation to physically/mathematically inclined readers. “veering” may be a better choice here.

Agreed. We now use “veering off”.

Lines 135-138: a very helpful explanation of the presence of elemental sulfur at concentrations well above its solubility.

Lines 146-147: This sentence implies that N₂ production will occur only at the top of the chemocline. Couldn't this occur throughout the chemocline wherever both NO₃⁻ and S₀ are present?

*True. We have changed this sentence to:
“Elemental sulfur, which is transported through eddy diffusion throughout the chemocline, may fuel further nitrate consumption via denitrification...”*

Lines 161-213: This section is very clearly written and makes a strong case for the potential importance of SUP05 and the cryptic sulfur cycle. A variety of important contextual information appears to have been added in this revision, and I appreciate the work done to temper some of the statements I thought were too strong in the previous versions.

Lines 214 ff: this rewritten section is very clear and compelling.

Lines 275-276: This seems like a reasonable assumption given how similar the energy yields of nitrate and oxygen respiration are, but it may be worth noting that parenthetically.

It would seem like a reasonable assumption, but as we note at Line 290, the biomass yields for denitrification may be lower.

Line 303 and Figure 4: I suggest adding letters or bars or some other visual indicator to show the significant difference noted in line 303.

We believe that the box-and-whiskers plot presentation style of the data provides an intuitive impression of the similarities and differences between the samples. Adding in extra bars or symbols would, in our eyes, clutter the diagram and require more explanatory text.

Figure 2: A panel showing O₂ concentration would provide useful context here, though I realize that it would be a seventh panel, complicating the layout.

We thought about this, and had such a panel in a very early pre-submission version, however, moved it out to save space. We do present the oxygen profiles in the Supplementary Figure 4. For the interested reader, the dissolved oxygen dynamics for this period of time are nicely illustrated in Thomsen et al., JGR 2015.

=====~~End of Review~~=====